# Time-Uniform Confidence Spheres for Means of Random Vectors

**Ben Chugg**                                                    *benchugg@cmu.edu*
**Hongjian Wang**                                               *hjnwang@cmu.edu*
**Aaditya Ramdas**                                              *aramdas@cmu.edu*
*Departments of Statistics and Machine Learning*
*Carnegie Mellon University*

**Reviewed on OpenReview:** *https://openreview.net/forum?id=2NSb3cJE03*

## Abstract

We study sequential mean estimation in $\mathbb{R}^d$. In particular, we derive time-uniform confidence spheres—*confidence sphere sequences* (CSSs)—which contain the mean of random vectors with high probability simultaneously across all sample sizes. Our results include a dimension-free CSS for log-concave random vectors, a dimension-free CSS for sub-Gaussian random vectors, and CSSs for sub-$\psi$ random vectors (which includes sub-gamma, sub-Poisson, and sub-exponential distributions). Many of our results are optimal. For sub-Gaussian distributions we also provide a CSS which tracks a time-varying mean, generalizing Robbins' mixture approach to the multivariate setting. Finally, we provide several CSSs for heavy-tailed random vectors (two moments only). Our bounds hold under a martingale assumption on the mean and do not require that the observations be iid. Our work is based on PAC-Bayesian theory and inspired by an approach of Catoni and Giulini.

## 1 Introduction

We consider the classical problem of estimating the mean of random vectors in nonparametric settings. Perhaps unlike traditional approaches however, we would like to perform this task sequentially. That is, we are interested providing estimates of the mean which hold uniformly across time as more data arrives. To formalize, let $(X_t)_{t \geqslant 1}$ be a stream of iid random vectors in $\mathcal{X} \subset \mathbb{R}^d$ with mean $\mu$ drawn from some unknown distribution $P$. Our goal is to construct a non-trivial sequence of sets, $(C_t)_{t \geqslant 1}$, $C_t \subset \mathbb{R}^d$, such that

$$P(\forall t \geqslant 1 : \mu \in C_t) \geqslant 1 - \alpha, \tag{1}$$

for some predetermined $\alpha \in (0, 1)$. In $\mathbb{R}^1$, such a sequence is often called a *confidence sequence (CS)*. We will use the more general term *confidence sphere sequence* (CSS) to highlight that we are working in higher dimensions, and because our sets $C_t$ are spherical as opposed to, say, ellipsoidal. CSs and CSSs are in contradistinction to confidence *intervals* (CIs)—the latter hold only at fixed times $t = n$ determined prior to receiving any samples, while the former hold *simultaneously* at all stopping times. Consequently, they are often described as *time-uniform* or as *anytime-valid*. Importantly, CSs and CSSs obviate the need to deploy union bounds to gain uniformity over the sample size. This is attractive for tasks in which data are continuously monitored during collection and which may call for data-dependent stopping times, two properties disallowed by fixed-time constructions such as confidence intervals (where it is known as p-hacking).

The study of CSs was pioneered by Darling, Robbins, Siegmund, and Lai in the decade from 1967-76 (Darling and Robbins, 1967; Lai, 1976). While further inquiry laid somewhat dormant for several decades afterwards, they have recently regained interest due to novel mathematical techniques and applicability to sequential decision-making. Indeed, CSs are used by several companies for A/B testing and adaptive experimentation.[1] We refer the reader to Howard et al. (2021); Waudby-Smith and Ramdas (2023) for more history and modern work on this topic.

---

[1]E.g., see Adobe (2023), Evidently (2023), and Netflix (Lindon et al., 2022).

Thus far, most work on CSs for the mean of random observations has been in the scalar setting (with some exceptions discussed below). However, estimating higher-dimensional means arises naturally in a wide variety of tasks in statistics and machine learning, including multi-armed bandits (Dani et al., 2008; Abbasi-Yadkori et al., 2011), regression (Lai and Wei, 1982; Vidaurre et al., 2013), covariance estimation (Kuchibhotla and Chakrabortty, 2022), stochastic optimization (Feldman et al., 2017), and anomaly detection (Rousseeuw and Hubert, 2018), to name a few. Advances in time-uniform confidence regions for estimators can therefore translate to sequentially-valid versions of such tasks.

In this work we construct CSSs by means of PAC-Bayesian analysis (introduced below), inspired by the original work of Catoni and Giulini (2017; 2018) which is in the fixed-time setting. We combine their insights with recent work which provides a general framework for understanding how and when PAC-Bayesian bounds can be made time-uniform (Chugg et al., 2023). The resulting synthesis yields a general method for constructing CSSs under a variety of distributional assumptions.

A concise summary of our results is given in Table 1 and their relationship to the existing literature is explored in Table 2. All bounds are closed-form and easily computable. In more detail, our contributions are:

1. **Sub-Gaussian bounds.** Theorem 2.2 provides a CSS for conditionally sub-Gaussian distributions. The width of the CSS depends on the trace and norm of the covariance matrices instead of the dimension of the ambient space ($d$). Such "dimension-free" bounds are desirable when the data are expected to lie on some low-dimensional manifold, i.e., when the effective dimension is less than $d$. Theorem 2.3 leverages Theorem 2.2 to provide a CSS whose width shrinks at the optimal iterated-logarithm rate. Theorem 2.16 provides a CSS when the mean changes with time, generalizing a result of Robbins (1970) to the multivariate setting. For technical reasons to be discussed later, this final result is not dimension-free.

2. **Log-concave bounds.** Theorem 2.8 provides a dimension-free CSS for distributions with log-concave densities (and more generally for distributions with bounded 1-Orlicz norm) which include Laplace, logistic, Dirichlet, gamma, and beta distributions. This extends a recent result of Zhivotovskiy (2024). Theorem 2.9 then extends Theorem 2.8 to provide a bound which shrinks at an iterated-logarithm rate.

3. **Sub-$\psi$ bounds.** While sub-Gaussianity and log-concavity are particularly popular distributional classes, there are many other light-tailed conditions one can consider. Theorem 2.12 provides a CSS for conditionally sub-$\psi$ random vectors for super-Gaussian $\psi$-functions (which includes sub-Poisson, sub-exponential, and sub-gamma distributions). A bound for general $\psi$-functions is given by Theorem A.3. The downside of considering such a general class is that these CSSs are not dimension-free. Theorems 2.14 and 2.15 deploy Theorem 2.12 to obtain iterated-logarithm rates for sub-gamma and sub-exponential distributions.

4. **Heavy-tailed bounds.** We provide several results for heavy-tailed distributions under only a second moment assumption. Theorems 3.2 and 3.5 provide two semi-empirical CSSs (in the sense of being adaptive to the observed values). Theorem 3.5 is tighter but holds only under a conditional symmetry assumption. Theorem 3.7 provides a time-uniform extension of the dimension-free Catoni-Giulini estimator (Catoni and Giulini, 2018) and Theorem 3.8 provides an iterated-logarithm version of Theorem 3.7. All bounds stated in this section are dimension-free.

## 1.1 Related work

Given its centrality to many statistical procedures, estimating the mean of random vectors is a well studied topic. For sub-Gaussian random vectors in the fixed-time setting, state-of-the-art concentration comes from Hsu et al. (2012). Our result has the same asymptotics (it is off by Hsu et al.'s result by an non-leading additive factor), but is time-uniform and handles martingale-dependence. We also improve on more classical bounds for isotropic sub-Gaussian random vectors obtained via covering arguments (e.g., Rigollet and Hütter, 2023, Theorem 1.19). Meanwhile, for heavy-tailed distributions, it is known that the sample mean (and weighted variants) have suboptimal performance (Catoni, 2012). This led to the development of alternatives such as median-of-means (Lugosi and Mendelson, 2019b; Minsker, 2015) and threshold-based

| Result | Dim | Condition | Dim-free? | Martingale? | CF? | LIL? | n opt? |
|---|---|---|---|---|---|---|---|
| Thm. E.2 | $d \geqslant 1$ | $\|X_t\|$ bounded | | ✓ | ✓ | | |
| Thm. 2.2 | $d \geqslant 1$ | $\Sigma_t$-sub-Gaussian | ✓ | ✓ | ✓ | | |
| Cor. 2.5 | $d \geqslant 1$ | $\Sigma$-sub-Gaussian | ✓ | ✓ | ✓ | | ✓ |
| Thm. 2.3 | $d \geqslant 1$ | $\Sigma$-sub-Gaussian | | ✓ | ✓ | ✓ | |
| Thm. 2.16 | $d \geqslant 1$ | $\sigma_t$-sub-Gaussian | | ✓ | ✓ | | |
| Thm. 2.8 | $d \geqslant 1$ | Log-concave/finite $\Phi_1$-norm | ✓ | ✓ | ✓ | | |
| Thm. 2.9 | $d \geqslant 1$ | Log-concave/finite $\Phi_1$-norm | ✓ | ✓ | ✓ | ✓ | |
| Cor. A.1 | $d \geqslant 1$ | Log-concave/finite $\Phi_1$-norm | ✓ | ✓ | ✓ | | ✓ |
| Thm. 2.12, A.3 | $d \geqslant 1$ | $\Sigma_t$-sub-$\psi$ | | ✓ | ✓ | | |
| Thm. 2.14 | $d \geqslant 1$ | $\Sigma$-sub-gamma | | ✓ | ✓ | ✓ | |
| Thm. 2.15 | $d \geqslant 1$ | $\Sigma$-sub-exponential | | ✓ | ✓ | ✓ | |
| Cor. 2.5 | $d \geqslant 1$ | $\Sigma$-sub-exponential | | ✓ | ✓ | | ✓ |
| Thm. 3.2 | $d \geqslant 1$ | $\mathbb{E}[\|X_t\| \mid \mathcal{F}_{t-1}] \leqslant v^2$ | ✓ | ✓ | ✓ | | |
| Cor. 3.3, 3.4 | $d = 1$ | $\mathbb{E}[X_t \mid \mathcal{F}_{t-1}] \leqslant v^2$ | | ✓ | ✓ | | |
| Thm. 3.5 | $d \geqslant 1$ | $\mathbb{E}[\|X_t\|^2 \mid \mathcal{F}_{t-1}] \leqslant v^2$, sym. | ✓ | ✓ | ✓ | | |
| Thm. 3.7 | $d \geqslant 1$ | $\mathbb{E}[\|X_t\|^2 \mid \mathcal{F}_{t-1}] \leqslant v^2$ | ✓ | ✓ | ✓ | | |
| Thm. 3.8 | $d \geqslant 1$ | $\mathbb{E}[\|X_t\|^2 \mid \mathcal{F}_{t-1}] \leqslant v^2$ | ✓ | ✓ | ✓ | ✓ | |

Table 1: A roadmap of the results in this work. $\Phi_1$ refers to the 1-Orlicz norm; see Section 2.2. "sym." stands for conditionally symmetric, meaning that the observations obey $X_t - \mu \mid \mathcal{F}_{t-1} \sim -(X_t - \mu) \mid \mathcal{F}_{t-1}$. "Dim-free" stands for dimension-free and "martingale" for martingale-dependence (i.e., Assumption 2). "CF" stands for closed-form, meaning the bound does not require numerical methods to solve. "LIL" refers to sequential bound with iterated logarithm rates and "$n$ opt" refers to bound that are optimized for a particular sample size $n$.

estimators (Catoni and Giulini, 2018), the latter of which we will make time-uniform in Theorem 3.7. We refer to Lugosi and Mendelson (2019a) for a survey on fixed-time mean estimation under heavy tails.

In the sequential setting, the theory of "self-normalized processes" (de la Peña et al., 2004; 2009) has been leveraged to give anytime-valid confidence ellipsoids under sub-Gaussian assumptions (Abbasi-Yadkori et al., 2011; Faury et al., 2020). This line of work is focused on bounding the self-normalized quantity $\|V_t^{-1/2} S_t\|$ for $S_t = \sum_{i=1}^{t} \epsilon_i X_i$ where $\epsilon_i$ is the scalar noise of an online regression model, $X_i$ are multivariate observations, and $V_t$ is a variance process. This setting is quite distinct from our own, which seeks non-self-normalized bounds and does not have both observations and noise components. Section 2.5 provides a brief discussion of how some of our bounds can be adapted to anisotropic distributions. This yields bounds in the Mahalanobis norm with respect to the variance but these results remain distinct from self-normalized bounds. Whitehouse et al. (2023) give a general treatment of time-uniform self-normalized bounds which extends beyond regression but it remains incomparable to our own work. Moreover, the results of Abbasi-Yadkori et al. (2011); Whitehouse et al. (2023) only apply to sufficiently light-tailed distributions, and not to the more general heavy-tailed settings we consider in Section 3.

Meanwhile, Manole and Ramdas (2023) provide anytime bounds on the mean of random vectors using reverse submartingales. Their bounds achieve the optimal iterated-logarithm rate of $\sqrt{\log \log(t)/t}$ (Darling and Robbins, 1967). Their results are most comparable to our results for isotropic sub-exponential distributions, as they do not hold under heavy tails and are not dimension-independent. We are also able to handle slightly more general distributional assumptions, as Manole and Ramdas (2023) work only with iid data. We will further discuss the relation to their work in Section 2, but suffice it to say here that we can employ a geometric "stitching method" (a common tool in sequential analysis), yielding a bound which matches this optimal rate and has smaller constants. As for anytime bounds under heavy-tails, Wang and Ramdas (2023) provide CSs for scalar random variables which hold under a finite $k$-th moment assumption for $k \in (1, 2]$.

Much closer to the spirit of our work are the papers of Zhivotovskiy (2024), Nakakita et al. (2024), and Giulini (2018). All three use the same variational principle that is at the heart of our method to study the concentration of singular values of light-tailed random matrices (Zhivotovskiy, 2024), the means of heavy-tailed random matrices (Nakakita et al., 2024), and to estimate the Gram operator in Hilbert spaces (Giulini,

| Condition | Our result | Comparison | [A] | [C] | [D] | Notes |
|---|---|---|:---:|:---:|:---:|---|
| sub-Gaussian | Cor. 2.5 | Hsu et al. (2012), Thm. 1 | ✓ | | ✓ | Off by a factor of $\leqslant 2$ |
| sub-Gaussian | Cor. 2.5 | Rigollet and Hütter (2023), Thm. 1.19 | ✓ | ✓ | ✓ | |
| sub-Gaussian | Thm. 2.16 | Waudby-Smith et al. (2024), Thm. 2.2 | | | | Non-asymp., holds in $\mathbb{R}^d$ |
| log-concave | Cor. A.1 | Zhivotovskiy (2024), Prop. 3 | ✓ | | ✓ | |
| sub-$\psi$ | Thm. 2.12 | Manole and Ramdas (2023), Cor. 23 | | | | Handles anisotropy |
| sub-exponential | Cor. A.1 | Maurer and Pontil (2021), Prop. 7 | ✓ | | ✓ | |
| sub-exponential | Thm. 2.15 | Manole and Ramdas (2023), Sec. 4.7 | | ✓ | ✓ | |
| finite 2nd moment | Thm. 3.7 | Catoni and Giulini (2018), Prop. 2.1 | ✓ | | ✓ | |
| finite 2nd moment | Thm. 3.7 | Lugosi and Mendelson (2019b), Thm. 1 | ✓ | | ✓ | Not sub-Gaussian, (*) |
| finite 2nd moment | Thm. 3.7 | Minsker (2015), Thm. 3.1 | ✓ | | ✓ | (*) |

Table 2: Comparison of our results with relevant bounds in the literature. More commentary on the relationship is provided in the text. Boxed letters indicated that our result improves over the existing result by adding [A]nytime-validity, improving the [C]onstants, or weakening the [D]ependence assumptions among the observations. (*) indicates that the estimators are built under different assumptions: we assume a bound on the raw second moment while they assume a bound on the centered second moment.

2018). While the underlying perspective is similar to our own, our goals are distinct. Moreover, these authors work in the fixed-time setting and none handle martingale dependence. Despite these differences, however, we believe that these three papers along with our own suggest that the variational approach to concentration is a promising research avenue.

Finally, recent work by Duchi and Haque (2024) shows that any fixed-time estimator can be converted into a sequential estimator with a loss of an iterated logarithm factor (i.e., $O(\log \log t)$) via a "doubling trick." Their construction employs a union bound over fixed-time estimators applied at times $t = 2^k$, $k \geqslant 1$. At times $j \in (2^k, 2^{k+1})$ one continues to use the estimator for time $2^k$. This is an insightful theoretical construction for obtaining optimal rates, and we compare our results with their bound in Sections 2.1 and 3.3. However, we note that it does have drawbacks. First, it is stated only for iid data. Second, it is not sequential in the practical sense that a data analyst would like. Indeed, if data collection stops at time $t = 2^k - 1$, then we must discard $2^k - 2^{k-1} - 1$ observations. The estimators proposed here can be updated after every time step, thus using all available data. This results in bounds that are often tighter than fixed-time bounds augmented with the doubling trick (see Figures 1 and 2).

### 1.2 Background and approach

Let us introduce some technical tools. A (forward) filtration $\mathcal{F} \equiv (\mathcal{F}_t)_{t \geqslant 0}$ is a sequence of $\sigma$-fields such that $\mathcal{F}_t \subset \mathcal{F}_{t+1}$, $t \geqslant 1$. In this work we always take $\mathcal{F}_t = \sigma(X_1, \ldots, X_t)$, the $\sigma$-field generated by the first $t$ observations. A stochastic process $S = (S_t)_{t \geqslant 1}$ is *adapted* to $(\mathcal{F}_t)$ if $S_t$ is $\mathcal{F}_t$ measurable for all $t$ (meaning that $S_t$ depends on $X_1, \ldots, X_t$), and *predictable* if $S_t$ is $\mathcal{F}_{t-1}$ measurable for all $t$ (meaning that $S_t$ depends only on $X_1, \ldots, X_{t-1}$). A *P-martingale* is an integrable stochastic process $M \equiv (M_t)_{t \geqslant 1}$ that is adapted to $\mathcal{F}$, such that $\mathbb{E}_P[M_{t+1}|\mathcal{F}_t] = M_t$ for all $t$. If '=' is replaced with '$\leqslant$', then $M$ is a *P-supermartingale*.

Our results rely on PAC-Bayesian theory which has emerged as successful method to bound the generalization gap of randomized predictors in statistical learning (Shawe-Taylor and Williamson, 1997; McAllester, 1998). As the name suggests, PAC-Bayesian bounds employ the "probably approximately correct" (PAC) learning framework of Valiant (1984) but addressed from a Bayesian perspective. Roughly speaking, rather than bounding the worst case risk, PAC-Bayesian bounds place with a prior over the parameter space $\Theta$, and provide uniform guarantees over all "posterior" distributions. Instead of containing a measure of the complexity of $\Theta$, the bounds involve a divergence term between the prior and posterior. We refer the unfamiliar reader to Guedj (2019) and Alquier (2024) for an introduction to such bounds and their applications.

For our purposes, the important feature of PAC-Bayesian bounds is the uniformity over posterior distributions. The insight of Catoni and Giulini (2018) is to translate this uniformity of distributions into a bound on $\langle \vartheta, \epsilon \rangle$ across all $\vartheta \in \mathbb{S}^{d-1}$, where $\epsilon$ is the error of the estimator. Such a bound translates immediately into a confidence sphere. Recently, uniformity over posteriors was extended to uniformity over posteriors *and time* by Chugg et al. (2023).

**Proposition 1.1** (Corollary of Theorem 4, Chugg et al., 2023). *For each $\theta \in \Theta$, assume that $Q(\theta) \equiv (Q_t(\theta))_{t \geqslant 1}$ is a nonnegative supermartingale with initial value 1. Consider a prior distribution $\nu$ over $\Theta$ (chosen before seeing the data). Then, with probability at least $1 - \alpha$, we have that simultaneously for all times $t \geqslant 1$ and posteriors $\rho \in \mathscr{M}(\Theta)$,*

$$\int_\Theta \log Q_t(\theta) \rho(\mathrm{d}\theta) \leqslant D_{\mathrm{KL}}(\rho \| \nu) + \log(1/\alpha). \tag{2}$$

Here, $\mathscr{M}(\Theta)$ is the set of distributions over $\Theta$ and $D_{\mathrm{KL}}(\cdot \| \cdot)$ is the KL divergence, which we recall is defined as $D_{\mathrm{KL}}(\rho \| \nu) = \int_\Theta \log\left(\frac{\mathrm{d}\rho}{\mathrm{d}\nu}\right) \mathrm{d}\rho$ if $\rho \ll \nu$ and $\infty$ otherwise. The uniformity over time and posteriors refers to the form $\mathbb{P}(\forall t, \forall \rho, \int_\Theta \log Q_t(\theta) \rho(\mathrm{d}\theta) \leqslant D_{\mathrm{KL}}(\rho \| \nu) + \log(1/\alpha)) \geqslant 1 - \alpha$ of the above bound, where the quantifiers are inside the probability.

At its core, our approach involves a judicious selection of $\Theta$, $Q$, family of posteriors $\{\rho\}$ and then applying Proposition 1.1. We will consider either uniform, truncated Gaussian, or Gaussian distributions over the parameter space $\Theta$, which we will take to be either a subset of $\mathbb{S}^{d-1} := \{x \in \mathbb{R}^d : \|x\| = 1\}$, $\mathbb{B}^d := \{x \in \mathbb{R}^d : \|x\| \leqslant 1\}$, or $\mathbb{R}^d$. The choice of $\Theta$ is driven by the availability of appropriate supermartingales over $\Theta$. We will typically choose supermartingales $Q_t(\theta)$ to be functions of $\sum_{i \leqslant t}\langle \theta, f_i(X_i) - \mu \rangle$ for some functions $f_i$, with the aim of obtaining bounds on $\sup_{\vartheta \in \mathbb{S}^{d-1}} \langle \vartheta, \sum_i (f_i(X_i) - \mu) \rangle = \|\sum_i (f_i(X_i) - \mu)\|_2$, which will furnish a CSS.

For example, for iid $X_1, X_2, \ldots$ which are $\Sigma$-sub-Gaussian, the process defined by $M_t(\theta) = \prod_{i \leqslant t} \exp\{\lambda \langle \theta, X_i - \mu \rangle - \frac{\lambda^2}{2} \langle \theta, \Sigma \theta \rangle\}$ for any $\lambda \geqslant 0$ is a nonnegative supermartingale for all $\theta \in \mathbb{R}^d$. Let $\rho_\theta$ be a Gaussian with mean $\theta$ and covariance $\beta^{-1} I_d$ for some scalar $\beta > 0$. Applying Proposition 1.1 to $(M_t(\theta))$ with $\Theta = \mathbb{R}^d$, prior $\nu = \rho_0$ and posteriors $\rho_\vartheta$ for all $\vartheta \in \mathbb{S}^{d-1}$, we obtain that $\mathbb{P}(\forall t \geqslant 1, \forall \vartheta \in \mathbb{S}^{d-1} : \sum_{i \leqslant t} \lambda \langle \vartheta, X_i - \mu \rangle \leqslant t \frac{\lambda^2}{2} \mathbb{E}_{\rho_\vartheta}\langle \theta, \Sigma \theta \rangle + \beta/2 + \log(1/\alpha)) \geqslant 1 - \alpha$, where we've calculated $D_{\mathrm{KL}}(\rho_\vartheta \| \nu) = \beta/2$. Using that $\sup_{\vartheta \in \mathbb{S}^{d-1}} \langle \vartheta, X_i - \mu \rangle = \|X_i - \mu\|$ and noting that $\mathbb{E}_{\rho_\vartheta}\langle \theta, \Sigma \theta \rangle = \langle \vartheta, \Sigma \vartheta \rangle + \mathrm{Tr}(\Sigma)/\beta \leqslant \|\Sigma\| + \mathrm{Tr}(\Sigma)/\beta$, we find that with probability $1 - \alpha$, for all $t \geqslant 1$,

$$\left\| \frac{1}{t} \sum_{i \leqslant t} X_i - \mu \right\| \leqslant \frac{\lambda}{2}\left(\|\Sigma\| + \frac{\mathrm{Tr}(\Sigma)}{\beta}\right) + \frac{\beta/2 + \log(1/\alpha)}{t\lambda}. \tag{3}$$

The parameters $\beta, \lambda > 0$ may then be chosen to optimize the width of the bound. Subsequent sections will expand on this example, relaxing both the distributional and dependence assumptions among the observations, and introducing various methods to ensure the width of the bound shrinks to zero over time (either via stitching or predictable plug-ins). Throughout, the norm $\|\cdot\|$ should be taken to be the $\ell^2$-norm $\|\cdot\|_2$ when applied to vectors, and the operator norm when applied to matrices.

## 1.3 Assumptions

The most common assumption in the literature is that the data are independently and identically distributed (iid). This is the case in most prior work on estimating means (cf. Catoni and Giulini, 2017; Lugosi and Mendelson, 2019b; Devroye et al., 2016; Joly et al., 2017).

**Assumption 1.** $X_1, X_2, \ldots$ are iid with mean $\mu$.

In this work, we are able to weaken this condition. We enforce only that the data stream has a constant *conditional* mean $\mu$. This allows for more "adversarial" distributions, and is particularly relevant in several bandit and online learning settings in which the iid assumption does not hold.

**Assumption 2.** $X_1, X_2, \ldots$ have constant conditional mean $\mathbb{E}[X_t | \mathcal{F}_{t-1}] = \mu$.

Of course, Assumption 1 implies Assumption 2. Assumption 2 is common in work on CSs, typically because they rely on (super)martingales which naturally enable such flexibility. We will often refer to Assumption 2 as *martingale-dependence*. While Abbasi-Yadkori et al. (2011) and Whitehouse et al. (2023) also work under more general conditions than iid data, we are—as far as we are aware—the first to do so in multivariate, heavy-tailed settings.

## 2 Light-tailed random vectors

### 2.1 Sub-Gaussian bounds

We say that $X_t$ is $\Sigma_t$-sub-Gaussian for some $\mathcal{F}_t$-measurable PSD matrix $\Sigma_t$ if

$$\mathbb{E}_P[\exp\{\lambda\langle\theta, X_t - \mu\rangle - \psi_N(\lambda)\langle\theta, \Sigma_t\theta\rangle\}|\mathcal{F}_{t-1}] \leqslant 1, \quad \text{for all } \theta \in \mathbb{R}^d \text{ and } \lambda \in \mathbb{R}, \tag{4}$$

where $\psi_N(\lambda) = \lambda^2/2$ and $\Sigma_t$ is PSD. The subscript "N" refers to the tail condition of a normal distribution and can be contrasted with other tail conditions investigated in Section 2.3. Unlike typical notions of sub-Gaussianity, (4) allows $(\Sigma_t)$ to be an $(\mathcal{F}_t)$-*adapted* sequence, not simply a predictable sequence. This allows $\Sigma_t$ to depend, for instance, on $X_t$. E.g., if $\mathbb{E}[\|X_t - \mu\|^2|\mathcal{F}_{t-1}] < \infty$, then we may take $\Sigma_t = \frac{1}{3}(\sum_{i\leqslant t}\|X_i - \mu\|^2 + \sum_{i\leqslant t}\mathbb{E}[\|X_i - \mu\|^2|\mathcal{F}_{i-1}]$ (see Lemma 3 part (f) in Howard et al. 2020). This example cannot be handled if $\Sigma_t$ must be $\mathcal{F}_{t-1}$-measurable.

We note that many authors impose isotropic conditions on the distribution when defining sub-Gaussianity, meaning that they take $\Sigma_t = \sigma_t^2 I_d$. However, allowing for anisotropy will enable us to give several dimension-free bounds that will depend on $\Sigma_t$ instead of $d$. As discussed in the introduction, such results are desirable when the data have low intrinsic dimension compared to the dimension of the ambient space. In Appendix A.1 we give the resulting bound when the distribution is isotropic (i.e., $\Sigma_t = \sigma_t^2 I_d$).

**Remark 2.1.** The inequality in (4) is often written as holding for all $\theta \in \mathbb{S}^{d-1}$ instead of $\theta \in \mathbb{R}^d$. The statements are, of course, equivalent (though they cease to be for different $\psi$-functions; see Section 2.3) but we wrote (4) as we did to highlight that we can take our parameter space as $\Theta = \mathbb{R}^d$. To elaborate, (4) naturally defines a supermartingale for each $\theta \in \mathbb{R}^d$. When applying Proposition 1.1, we can thus use Gaussians as the prior and posterior distributions as opposed to restricting ourselves to uniform distributions or truncated Gaussians as in Sections 2.2 and 2.3.

Let $\rho_\vartheta$ be a Gaussian with mean $\vartheta$ and covariance $\beta^{-1}I_d$ for some real $\beta > 0$. The process defined by $M_t(\theta) = \prod_{i=1}^t \exp\{\lambda_i\langle\theta, X_i - \mu\rangle - \psi_N(\lambda_i)\langle\theta, \Sigma_i\theta\rangle\}$ is a nonnegative $P$-supermartingale for all $\theta \in \mathbb{R}^d$ by (4). Applying Proposition 1.1 to this family of processes with prior $\rho_0$ and posteriors $\rho_\vartheta$ for $\vartheta \in \mathbb{S}^{d-1}$, one obtains the following result. The details may be found in Appendix C.1.

**Theorem 2.2.** *Let $(X_t)_{t\geqslant 1}$ be conditionally $(\Sigma_t)$-sub-Gaussian and satisfy Assumption 2. Let $(\lambda_t)$ be a predictable sequence in $(0,\infty)$ and fix $\beta > 0$. Then, with probability $1 - \alpha$, simultaneously for all $t \geqslant 1$,*

$$\left\|\frac{\sum_{i\leqslant t}\lambda_i X_i}{\sum_{i\leqslant t}\lambda_i} - \mu\right\| \leqslant \frac{\sum_{i\leqslant t}\psi_N(\lambda_i)(\|\Sigma_i\| + \beta^{-1}\operatorname{Tr}(\Sigma_i)) + \beta/2 + \log(1/\alpha)}{\sum_{i\leqslant t}\lambda_i}.$$

Let us turn straightaway to the question of how to choose $\beta$ and $\lambda_t$ above. If $\sup_t\|\Sigma_t\| \leqslant \|\Sigma\| < \infty$ and $\sup_t\operatorname{Tr}(\Sigma_t) \leqslant \operatorname{Tr}(\Sigma) < \infty$, then consider taking

$$\beta = \sqrt{\frac{2\operatorname{Tr}(\Sigma)\log(1/\alpha)}{\|\Sigma\|}} \quad \text{and} \quad \lambda_t = \sqrt{\frac{\beta + 2\log(1/\alpha)}{f_\beta(\Sigma)t\log(t+1)}} \tag{5}$$

where $f_\beta(\Sigma) = \|\Sigma\| + \operatorname{Tr}(\Sigma)/\beta$. Using that $\sum_{i\leqslant t}(i\log(i+1))^{-1/2} \asymp \sqrt{t/\log(t)}$ and $\sum_{i\leqslant t}(i\log(i+1))^{-1} \asymp \log\log(t)$, we see that the width of the CSS in Theorem 2.2 is

$$\widetilde{O}\left(\sqrt{\operatorname{Tr}(\Sigma) + \|\Sigma\|\log(1/\alpha)}\sqrt{\frac{\log t}{t}}\right). \tag{6}$$

where $\widetilde{O}$ hides iterated logarithm factors. If $\sup_t\|\Sigma_t\|$ or $\sup_t\operatorname{Tr}(\Sigma_t)$ are unbounded, then we cannot guarantee that the width will shrink to zero. We can obtain iterated logarithm rates (i.e., $\log\log t$ instead of $\log t$) in (6) with a technique known as stitching (Howard et al., 2021). This is similar to the doubling technique of Duchi and Haque (2024), but relies on applying Theorem 2.2 in each epoch, thus resulting an estimator which is updated at every timestep, and not just at times $2^k$. Details may be found in Appendix B. Note that Theorem 2.3 posits that $\Sigma_t = \Sigma$ for all $t$, thus assuming that $\Sigma$ is non-random.

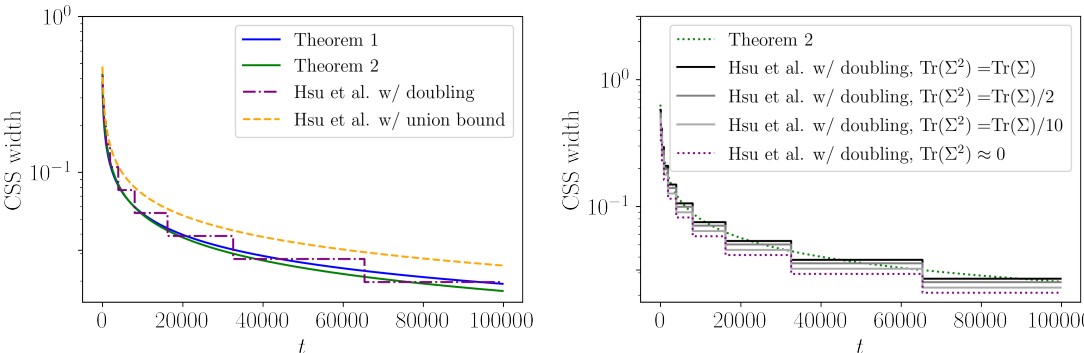

Figure 1: **Left:** Comparison of Theorem 2.2 and its stitched version, Theorem 2.3, against the results of Hsu et al. (2012). The latter is made time-uniform in two ways: by a union bound (dotted orange line) and by the doubling technique of Duchi and Haque (2024) (dotted purple). We begin the plotting the width at $t = 150$ for scale purposes. We fix $\|\Sigma\| = 1$ and take $\mathrm{Tr}(\Sigma) = 5$. **Right:** A comparison of estimators with iterated logarithm rates. We plot Theorem 2.3 against the bound of Hsu et al.—given iterated logarithm rates via Duchi and Haque (2024)—for various ratios of $\mathrm{Tr}(\Sigma^2)$ to $\mathrm{Tr}(\Sigma)$. As $\mathrm{Tr}(\Sigma^2)$ shrinks Theorem 2.3 starts to be dominated by the bound of Hsu et al.. Simulation details can be found in Appendix D.

**Theorem 2.3.** *Let $(X_t)_{t \geqslant 1}$ be conditionally $\Sigma$-sub-Gaussian and satisfy Assumption 2. Then, with probability $1 - \alpha$, simultaneously for all $t \geqslant 1$,*

$$\left\| \frac{1}{t} \sum_{i \leqslant t} X_i - \mu \right\| \leqslant 1.21 \sqrt{\frac{\mathrm{Tr}(\Sigma)}{t}} + 1.682 \sqrt{\frac{\|\Sigma\|(\log(1.65/\alpha) + 2\log(\log_2(t) + 1))}{t}}. \tag{7}$$

Figure 1 plots the width of the CSSs in Theorems 2.2 and 2.3. We compare our results to the state-of-the-art fixed-time result of Hsu et al. (2012) (described below; see (9)), which we make time-uniform in two ways: via a naive union bound, and by the doubling technique of Duchi and Haque (2024). As expected, a naive union bound is outperformed by all other bounds. Theorem 2.3 uniformly dominates Theorem 2.2 at all time steps. Neither Hsu et al. with doubling nor Theorem 2.3 uniformly dominate the other, except for small values $\mathrm{Tr}(\Sigma^2)/\mathrm{Tr}(\Sigma)$, in which case the former begins to dominate. Note however that $\mathrm{Tr}(\Sigma^2) \geqslant \frac{2}{1+\sqrt{d}} \mathrm{Tr}(\Sigma)\|\Sigma\|$ (see Lemma C.1 in Appendix C.1), so that values of $\mathrm{Tr}(\Sigma^2)/\mathrm{Tr}(\Sigma)$ close to zero require large dimension.

**Remark 2.4.** A dissatisfying aspect of Theorem 2.2 result is that $\beta$ is fixed while $\Sigma_t$ can vary. Ideally, one could allow $\beta = \beta_t$ to be a function of $t$ and take $\beta_t \asymp \sqrt{\mathrm{Tr}(\Sigma_t)}$. In fact, such a choice is possible if $\Sigma_t$ is predictable—the general version of Proposition 1.1 stated in Chugg et al. (2023) allows us to choose predictable posteriors. However, since the covariance of the prior is fixed, this strategy results in a factor of $d$ in the KL-divergence. Our bounds are therefore most useful when upper bounds on both $\mathrm{Tr}(\Sigma_t)$ and $\|\Sigma_t\|$ across all $t$ are known, in which case $\beta$ and $\lambda_t$ can be optimized as above.

**Fixed-time optimization.** Next, let us instantiate Theorem 2.2 optimized for a specific time $t = n$ with $\Sigma_t = \Sigma$ for all $t$. Some elementary calculus suggests that setting

$$\lambda_i = \lambda := \sqrt{\frac{\beta + 2\log(1/\alpha)}{n(\mathrm{Tr}(\Sigma)\beta^{-1} + \|\Sigma\|)}}, \quad \beta := \sqrt{\frac{2\mathrm{Tr}(\Sigma)\log(1/\alpha)}{\|\Sigma\|}}.$$

results in the optimal width of the boundary. This gives:

**Corollary 2.5.** *Let $(X_t)_{t \geqslant 1}$ be $\Sigma$-sub-Gaussian and satisfy Assumption 2. Fix some $n \in \mathbb{N}$. Then, with probability $1 - \alpha$, simultaneously for all $t \geqslant 1$,*

$$\left\| \frac{1}{t} \sum_{i=1}^{t} X_i - \mu \right\| \leqslant \left( \frac{1}{2\sqrt{n}} + \frac{\sqrt{n}}{2t} \right) \left( \sqrt{\mathrm{Tr}(\Sigma)} + \sqrt{2\|\Sigma\|\log(1/\alpha)} \right). \tag{8}$$

Corollary 2.5 may strike the reader as somewhat odd at first glance. The bound is optimized for a particular sample size $n$ but remains valid at all times due to the time-uniformity of Theorem 2.2. However, the bound is tightest at $t = n$ and the width does not to go zero as $t \to \infty$. Thus while it is a time-uniform bound, it is most useful at sample sizes that are close to $n$. It might be compared to Freedman-style inequalities (Freedman, 1975) (which hold for all $t \leqslant n$), or de la Pẽna style inequalities (de la Pẽna, 1999) (which hold for all $t \geqslant n$ but also do not shrink to zero).

Hsu et al. (2012) prove that for iid $\Sigma$-sub-Gaussian random vectors $X_1, \ldots, X_n$, with probability $1 - \alpha$,

$$\left\| \frac{1}{n} \sum_{i \leqslant n} X_i - \mu \right\| \leqslant \sqrt{\frac{\operatorname{Tr}(\Sigma) + 2\|\Sigma\| \log(1/\alpha) + 2\sqrt{\operatorname{Tr}(\Sigma^2) \log(1/\alpha)}}{n}}. \tag{9}$$

Instantiating Corollary 2.5 at $t = n$, we see that the width of (8) differs from (9) by replacing $\sqrt{2 \operatorname{Tr}(\Sigma) \|\Sigma\| \log(1/\alpha)}$ with $2\sqrt{\operatorname{Tr}(\Sigma^2) \log(1/\alpha)}$ (this is clear after writing $\sqrt{\operatorname{Tr}(\Sigma)} + \sqrt{2\|\Sigma\| \log(1/\alpha)} = \sqrt{\operatorname{Tr}(\Sigma) + 2\|\Sigma\| \log(1/\alpha) + 2\sqrt{2 \operatorname{Tr}(\Sigma) \|\Sigma\| \log(1/\alpha)}}$). The bound in (8) is therefore somewhat slightly looser than (9) at $t = n$, but the bounds differ by at most a factor of two as demonstrated by the following lemma. Despite this looseness, we emphasize that our bound holds under martingale dependence, is sequential, and performs better than a sequentialized version of (9) in many regimes (see Figure 1).

**Lemma 2.6.** *Let $W_n$ be the right hand side of (8) at $t = n$ and let $H_n$ be the right hand side of (9). Then $W_n/H_n \leqslant 2$.*

*Proof.* We have $\sqrt{n} W_n = \sqrt{\operatorname{Tr}(\Sigma)} + \sqrt{2\|\Sigma\| \log(1/\alpha)} \leqslant 2 \max\{\sqrt{\operatorname{Tr}(\Sigma)}, \sqrt{2\|\Sigma\| \log(1/\alpha)}\}$. Further, note that $\sqrt{n} H_n = \sqrt{\operatorname{Tr}(\Sigma) + 2\|\Sigma\| \log(1/\alpha) + 2\sqrt{\operatorname{Tr}(\Sigma^2) \log(1/\alpha)}} \geqslant \sqrt{\operatorname{Tr}(\Sigma) + 2\|\Sigma\| \log(1/\alpha)}$. Using that $\max\{\sqrt{a}, \sqrt{b}\} \leqslant \sqrt{a + b}$ completes the argument. $\square$

When $\Sigma = \sigma^2 I$, Corollary 2.5 improves the constants in a classical concentration result for sub-Gaussian random vectors, which states that, with probability $1 - \alpha$, $\left\| \frac{1}{n} \sum_i X_i - \mu \right\| \leqslant 4\sigma\sqrt{d}/\sqrt{n} + 2\sigma\sqrt{\log(1/\alpha)}/\sqrt{n}$ (Rigollet and Hütter, 2023, Theorem 1.19). We also remark that Jin et al. (2019) study concentration under the weaker assumption of so-called *norm*-sub-Gaussianity. When their result is translated into our setting, one obtains that with probability $1 - \alpha$, $\left\| \frac{1}{n} \sum_i X_i - \mu \right\| = O(\sigma\sqrt{d \log(2d/\alpha)/n})$. This is worse than Corollary 2.5, though in general the two bounds are incomparable, holding for different assumptions.

### 2.2 Log-concave distributions and finite Orlicz norm

If $P$ is a distribution whose density $f(x)$ can be written as $f(x) = \exp(\phi(x))$ for some concave function $\phi$ then we say that $P$ is a *log-concave* distribution. Many popular distributions are log-concave: the Gaussian, Laplace, logistic, uniform, Dirichlet, gamma, and beta distributions being several examples. Here we extend a recent observation of Zhivotovskiy (2024) and demonstrate that all log-concave distributions admit a dimension-free CSS.

In fact, we deal with distributions slightly more general than log-concave distributions. We consider sequences of observations $(X_t)_{t \geqslant 1}$ with a finite conditional $\Phi_1$-Orlicz norm:

$$\| \langle \theta, X_t - \mathbb{E}_P[X_t | \mathcal{F}_{t-1}] \rangle \|_{\Phi_1} \leqslant C \sqrt{\langle \theta, \Sigma \theta \rangle}, \quad \text{for all } \theta \in \mathbb{R}^d, \tag{10}$$

where we recall that for a scalar-valued random variable $Y$,

$$\|Y\|_{\Phi_1} = \inf \left\{ \epsilon > 0 : \mathbb{E}_P \exp\left( \frac{|Y|}{\epsilon} \right) \leqslant 2 \right\}.$$

In (10), $\Sigma$ is assumed to be PSD and deterministic (unlike the sub-Gaussian case). If $P$ is log-concave, then Adamczak et al. (2010, Lemma 2.3) demonstrates that it obeys (10).[2] For Gaussians, the constant is $C = 2$ (in fact, $C = 2$ for all symmetric distributions, see Adamczak et al. 2010). Sub-exponential distributions

---

[2]They state the result for all $\theta \in \mathbb{S}^{d-1}$, but this is easily seen to be equivalent to (10).

also satisfy (10). Indeed, (10) is another way to define sub-exponential distributions, and the results in this section thus provide dimension-free CSSs for such distributions. However, for *isotropic* sub-exponential distributions, the results are sub-optimal compared to those in the next section (Theorem 2.15 in particular). We will comment more on the relationship in Section 2.3.

Unlike sub-Gaussian distributions, the log-concave condition does not immediately imply a supermartingale which we can deploy in Proposition 1.1. Instead we consider a process which has the form $\prod_{i \leqslant t} \exp\{g_i(\theta) - \log \mathbb{E} \exp g_i(\theta)\}$ which is a martingale whenever $\log \mathbb{E}[\exp(g_i(\theta))]$ is finite. In order to bound this log-mgf term, we appeal to the following lemma, which is standard but proved explicitly by Zhivotovskiy (2024, Lemma 2) (see also Howard et al. 2020, Proposition 1).

**Lemma 2.7.** *For a scalar random variable $Y$ such that $\|Y - \mathbb{E}Y\|_{\Phi_1} < \infty$, we have*

$$\mathbb{E} \exp(\lambda(Y - \mathbb{E}Y)) \leqslant \exp\Big(4\lambda^2 \|Y - \mathbb{E}Y\|_{\Phi_1}^2\Big), \quad for \ all \ \ |\lambda| \leqslant \frac{1}{2\|Y - \mathbb{E}Y\|_{\Phi_1}}.$$

We deploy Lemma 2.7 with $Y = g_i(\theta)$. It's tempting to consider $g_i(\theta) = \lambda_i \langle \theta, X_i - \mu \rangle$ and to mimic the approach used in the sub-Gaussian case. Unfortunately, this strategy breaks down because $\|\langle \theta, X_t - \mu \rangle\|_{\Phi_1}$ may be arbitrarily large for certain choices of $\theta$, thus rendering Lemma 2.7 inapplicable for any $\lambda$. Therefore, we cannot use Gaussian distributions in Proposition 1.1 as was done in the sub-Gaussian case. Instead, following an approach used by Zhivotovskiy (2024) in the case of random matrices, we use a *truncated* Gaussian with a well-chosen radius. This, combined with a clever choice of $g_i(\theta)$ allows us to provide the following dimension-free bound. The details may be found in Appendix C.1. Let us define the function $h_\Sigma(u) := \sqrt{\text{Tr}(\Sigma)u} + u\sqrt{\|\Sigma\|}$ for $u > 0$.

**Theorem 2.8.** *Let $(X_t)_{t \geqslant 1}$ satisfy (10) and Assumption 2. Let $(\lambda_t)$ be a predictable sequence taking values in $(0, 1)$. Then with probability $1 - \alpha$, simultaneously for all $t \geqslant 1$,*

$$\left\| \frac{\sum_{i \leqslant t} \lambda_i X_i}{\sum_{i \leqslant t} \lambda_i} - \mu \right\| \leqslant \frac{2Ch_\Sigma(1)\sum_{i \leqslant t}\lambda_i^2 + 4Ch_\Sigma(\log(2/\alpha))}{\sum_{i \leqslant t}\lambda_i}. \tag{11}$$

Note that while log-concavity implies that some such $C$ exists, the bounds are unusable without knowledge of $C$. This is similar to needing to know the sub-Gaussian constant or variance bound in order to be able to construct finite-sample-valid confidence intervals.

Noting that $h_\Sigma(\cdot)$ is monotone, our bound recovers the one by Zhivotovskiy (2024) at $t = 1$ with $\lambda_1 = 1$: $2Ch_\Sigma(1) + 4Ch_\Sigma(\log(2/\alpha)) \leqslant 6Ch_\Sigma(\log(2/\alpha))$ assuming $\alpha \leqslant 2e^{-1}$. Further, following arguments in that work, this can be seen to be optimal in terms of dependence on $\log(1/\alpha), \text{Tr}(\Sigma)$, and $\|\Sigma\|$. Theorem 2.8 is also related to Maurer and Pontil (2021, Proposition 7.1), which uses entropy methods to prove that $\|\frac{1}{n}\sum_i X_i - \mu\| \leqslant 8e\|\|X\|_2\|_{\Phi_1}\sqrt{\log(2/\delta)/n}$ for iid $X_1, \ldots, X_n$. One can show that $\|\|X\|_2\|_{\Phi_1} \lesssim \sqrt{\text{Tr}(\Sigma)}$, so this bound is of the same order as ours, the width of which we now analyze.

Consider

$$\lambda_t \asymp \sqrt{\frac{h_\Sigma(u_\alpha)}{h_\Sigma(1)t\log(t+1)}},$$

where $u_\alpha := \log(2/\alpha)$. If $\alpha \leqslant 2e^{-1}$, then

$$h_\Sigma(u_\alpha)h_\Sigma(1) = (\sqrt{\text{Tr}(\Sigma)u_\alpha} + u_\alpha\sqrt{\|\Sigma\|})(\sqrt{\text{Tr}(\Sigma)} + \sqrt{\|\Sigma\|})$$

$$= \text{Tr}(\Sigma)\sqrt{u_\alpha} + (u_\alpha + \sqrt{u_\alpha})\sqrt{\text{Tr}(\Sigma)\|\Sigma\|} + u_\alpha\|\Sigma\|$$

$$\leqslant \text{Tr}(\Sigma)\sqrt{u_\alpha} + 3u_\alpha\sqrt{\text{Tr}(\Sigma)\|\Sigma\|},$$

where the inequality follows from $u_\alpha \geqslant 1$ and $\text{Tr}(\Sigma) \geqslant \|\Sigma\|$. With these choices, the CSS in Theorem 2.8 has width

$$\widetilde{O}\left(C\sqrt{(\text{Tr}(\Sigma)\sqrt{u_\alpha} + u_\alpha\sqrt{\text{Tr}(\Sigma)\|\Sigma\|})}\sqrt{\frac{\log t}{t}}\right), \tag{12}$$

where $C$ is the log-concave constant. Here we've once again used the fact that $\sum_{i\leqslant t}(i\log(i+1))^{-1} \asymp \log\log t$ and $\sum_{i\leqslant t}(\sqrt{i\log(i+1)})^{-1/2} \asymp \sqrt{t\log t}$. As in the sub-Gaussian case, we can also use stitching to provide a bound which shrinks at the optimal iterated logarithm rate. The details may be found in Appendix B.

**Theorem 2.9.** *Let* $(X_t)_{t\geqslant 1}$ *satisfy* (10) *and Assumption 2. For any* $\alpha \in (0,1)$ *and* $t \geqslant 1$*, set* $r_\alpha(t) := 2\log(\log_2(t)+1)+\log(3.3/\alpha)$*. Then, with probability* $1-\alpha$*, simultaneously for all* $t$ *such that* $t \geqslant 2\sqrt{\frac{h_\Sigma(r_\alpha(t))}{h_\Sigma(1)}}$*,*

$$\left\|\frac{\sum_{i\leqslant t}\lambda_i X_i}{\sum_{i\leqslant t}\lambda_i} - \mu\right\| \leqslant 6.73C\sqrt{\frac{h_\Sigma(1)h_\Sigma(r_\alpha(t))}{t}} \leqslant 6.73C\sqrt{\frac{\mathrm{Tr}(\Sigma)\sqrt{r_\alpha(t)}+3r_\alpha(t)\sqrt{\mathrm{Tr}(\Sigma)\|\Sigma\|}}{t}}. \quad (13)$$

### 2.3 Sub-$\psi$ distributions

Other light-tailed conditions on random vectors can be captured by assuming the log-MGF is bounded by some CGF-like function. More precisely, we assume that there exists a function $\psi : [0, \lambda_{\max}) \to \mathbb{R}$ for some $\lambda_{\max} \in (0, \infty]$ such that

$$\mathbb{E}_P[\exp\{\lambda\langle v, X_t - \mu\rangle\} - \psi(\lambda)\langle v, \Sigma_t v\rangle | \mathcal{F}_{t-1}] \leqslant 1 \quad \text{for all } v \in \mathbb{S}^{d-1}, \quad (14)$$

and all $\lambda \in [0, \lambda_{\max})$. Condition (14) is called a sub-$\psi$ process by Howard et al. (2020) and Whitehouse et al. (2023). As in the sub-Gaussian case, we assume that $\Sigma_t$ is PSD and $\mathcal{F}_t$-measurable (i.e., $(\Sigma_t)$ is $(\mathcal{F}_t)$-adapted not only predictable). Different choices of $\psi$ recover common distributions (terminology is borrowed from Howard et al. (2020); we take $1/0 = \infty$):

1. $\psi_N(\lambda) := \lambda^2/2$ for $\lambda \in [0, \lambda_{\max})$ results in a sub-exponential distribution (sub-Gaussian if $\lambda_{\max} = \infty$). In this case we say that $(X_t)$ is $(\Sigma_t, \lambda_{\max})$-sub-exponential.

2. $\psi_{G,c}(\lambda) = \frac{\lambda^2}{2(1-c\lambda)}$ for $c \in \mathbb{R}$ and $\lambda_{\max} = 1/\max\{c, 0\}$ results in a sub-gamma distribution (in the sense that $\psi_{G,c}$ is an *upper bound* on the CGF of a gamma random variable). In this case we say that $(X_t)$ is $(\Sigma_t, c)$-sub-gamma.

3. $\psi_{P,c}(\lambda) = (e^{c\lambda} - c\lambda - 1)/c^2$ for $c \in \mathbb{R}$ and $\lambda_{\max} = \infty$ results in a sub-Poisson distribution (in the sense that $\psi_{P,c}$ is the CGF of a centered unit-rate Poisson random variable). In this case we say that $(X_t)$ is $(\Sigma_t, c)$-sub-Poisson.

4. $\psi_{E,c}(\lambda) = (-\log(1 - c\lambda) - c\lambda)/c^2$ for $c \in \mathbb{R}$ where $\lambda_{\max} = 1/\max\{c, 0\}$ results in a sub-neg-exponential distribution ($\psi_{E,c}$ is the CGF of a centered unit-rate negative-exponential random variable).[3] In this case we say that $(X_t)$ is $(\Sigma_t, c)$-sub-neg-exponential.

In general, to indicate that $(X_t)$ satisfies (14) for some $\psi$-function, we will say it is $(\Sigma_t)$-sub-$\psi$. The five $\psi$-functions are all (upper bounds on) the CGF of various random variables. Further, all are (a) twice differentiable and (b) obey $\psi(0) = \lim_{x\to 0^+} \psi'(x) = 0$. We will thus call any function which satisfies (a) and (b) *CGF-like*. In what follows we will mainly focus on CGF-like functions that are *super-Gaussian* (Whitehouse et al., 2023). Formally, we say $\psi$ is super-Gaussian if $\lambda \mapsto \psi(\lambda)/\lambda^2$ is nondecreasing. Informally, this implies that $\psi$ grows at least as quickly as the CGF of a standard normal random variable. All of the $\psi$-functions enumerated above are super-Gaussian.

When restricting our attention to super-Gaussian CGF-like functions, (14) holds for all $v$ in the unit *ball* $\mathbb{B}^d$, as opposed to only the unit *sphere* $\mathbb{S}^{d-1}$—see Lemma 2.10. This allows us to use uniform distributions over the ball in Proposition 1.1. For $\psi$ functions that are not super-Gaussian, we can instead use the von Mises-Fisher distribution (Mardia et al., 2000) over the sphere. This, however, results in bounds with slightly worse dimension dependence and worse constants. Since most $\psi$ functions of interest are super-Gaussian, the result for non-super-Gaussian $\psi$ is given in the Appendix; see Theorem A.3.

**Lemma 2.10.** *Let* $(X_t)$ *be* $(\Sigma_t)$-sub-$\psi$ *where* $\psi$ *is super-Gaussian. Then* (14) *holds for all* $v \in \mathbb{B}^d$*.*

---

[3] Many authors call $\psi_{E,c}$ the CGF of an exponential distribution (Howard et al., 2020; 2021; Whitehouse et al., 2023), but we want to reserve the term sub-exponential for $\psi_N$ with finite $\lambda_{\max}$. Hence we use the term negative-exponential instead.

*Proof.* Let $\psi$ be super-Gaussian. Observe that for any $\theta \in \mathbb{B}^d$,

$$\frac{\psi(\lambda\|\theta\|)}{\lambda^2\|\theta\|^2} \leqslant \frac{\psi(\lambda)}{\lambda^2},$$

therefore $\psi(\lambda\|\theta\|) \leqslant \|\theta\|^2 \psi(\lambda)$. Consequently, letting $\vartheta = \theta/\|\theta\| \in \mathbb{S}^{d-1}$,

$$\begin{aligned}
&\exp\{\lambda_i\langle\theta, X_i - \mu\rangle - \psi(\lambda_i)\langle\theta, \Sigma_i\theta\rangle\} \\
&= \exp\{\lambda_i\|\theta\|\langle\vartheta, X_i - \mu\rangle - \psi(\lambda_i)\|\theta\|^2\langle\vartheta, \Sigma_i\vartheta\rangle\} \\
&\leqslant \exp\{\lambda_i\|\theta\|\langle\vartheta, X_i - \mu\rangle - \psi(\lambda_i\|\theta\|)\langle\vartheta, \Sigma_i\vartheta\rangle\}.
\end{aligned}$$

Since $\lambda_i\|\theta\| \leqslant \lambda_i \leqslant \lambda_{\max}$, it follows that $M_t(\theta)$ is upper bounded by

$$\overline{M}_t(\vartheta) = \prod_{i\leqslant t} \exp\{\lambda_i\|\theta\|\langle\vartheta, X_i - \mu\rangle - \psi(\lambda_i\|\theta\|)\langle\vartheta, \Sigma_i\vartheta\rangle\},$$

which defines a nonnegative supermartingale by (14), proving the claim. $\square$

**Remark 2.11.** While (14) superficially resembles the sub-Gaussian condition (4) for $\psi = \psi_N$, (14) holds only for all vectors in the unit sphere. The natural isotropic condition for general $\psi$ is $\sup_{v\in\mathbb{S}^{d-1}} \mathbb{E}_P[\exp\{\lambda\langle v, X_t - \mu\rangle|\mathcal{F}_{t-1}\}] \leqslant \exp\{\psi(\lambda)\}$. When $\lambda_{\max} = \infty$ then such a definition naturally extends to all vectors in $\mathbb{R}^d$ and is equivalent to (the isotropic version of) (4). For finite $\lambda_{\max}$, however, such an extension does not hold, so we work with (14) instead. This necessitates using different distributions in Proposition 1.1.

Suppose $\psi$ is super-Gaussian. Let $\rho_\vartheta$ be the uniform distribution over the ball centered at $\vartheta \in (1-\epsilon)\mathbb{S}^{d-1}$ with radius $\epsilon$. Let $\nu$ be the uniform distribution over the ball centered at 0 with radius 1. Observe that $\rho_\vartheta \ll \nu$. The KL-divergence between $\rho_\vartheta$ and $\nu$ is

$$\begin{aligned}
D_{\mathrm{KL}}(\rho_\vartheta\|\nu) &= \int_{\epsilon\mathbb{B}^d+\vartheta} \log\left(\frac{\mathrm{d}\rho_\vartheta}{\mathrm{d}\nu}(\theta)\right)\rho_\vartheta(\mathrm{d}\theta) \\
&= \int_{\epsilon\mathbb{B}^d+\vartheta} \log\left(\frac{\mathrm{vol}(\mathbb{B}^d)}{\mathrm{vol}(\epsilon\mathbb{B}^d)}\right)\rho_\vartheta(\mathrm{d}\theta) = d\log\left(\frac{1}{\epsilon}\right),
\end{aligned} \tag{15}$$

where $\mathrm{vol}(A)$ is the volume of a set $A \subset \mathbb{R}^d$, whence $\mathrm{vol}(\mathbb{B}^d) = \pi^{d/2}/\Gamma(d/2+1)$ and $\mathrm{vol}(\epsilon\mathbb{B}^d) = \pi^{d/2}\epsilon^d/\Gamma(d/2+1)$. We note that using uniform distributions in conjunction with PAC-Bayes arguments was also used by Lee et al. (2024) when developing confidence sequences for GLMs. Applying Proposition 1.1 to the family of processes $(M_t(\theta))_{t\geqslant 1}$ defined by

$$M_t(\theta) = \prod_{i=1}^t \exp\{\lambda_i\langle\theta, X_i - \mu\rangle - \psi(\lambda_i)\langle\theta, \Sigma_i\theta\rangle\}, \tag{16}$$

each of which is a nonnegative $P$-supermartingale by (14), gives the following result. The proof may be found in Appendix C.1. The closest result to the following bound is Manole and Ramdas (2023, Corollary 23), which gives a bound for isotropic sub-$\psi$ processes. Their result is stated in terms of the inverse of the convex conjugate of $\psi$, $(\psi^*)^{-1}$, which makes a direct comparison of the bounds challenging. However, for common $\psi$ functions, $(\psi^*)^{-1}(x)$ behaves as $\sqrt{x}$ for small $x$, making their result consistent with Theorem 2.14 below.

**Theorem 2.12.** *Let $(X_t)_{t\geqslant 1}$ be $\Sigma_t$-sub-$\psi$ and satisfy Assumption 2. Suppose $\psi$ is super-Gaussian. Let $(\lambda_t)$ be a predictable sequence in $(0, \lambda_{\max})^\mathbb{N}$. Fix $0 < \epsilon < 1$. Then, with probability $1-\alpha$, for all $t \geqslant 1$,*

$$\left\|\frac{\sum_{i\leqslant t}\lambda_i X_i}{\sum_{i\leqslant t}\lambda_i} - \mu\right\| \leqslant \frac{\sum_{i\leqslant t}\psi(\lambda_i)\|\Sigma_i\| + d\log(1/\epsilon) + \log(1/\alpha)}{(1-\epsilon)\sum_i\lambda_i}. \tag{17}$$

A reasonable default value for $\epsilon$ is $1/2$, but it may be further optimized depending on the relative size of $d$, $\alpha$, and $\sup_t\|\Sigma_t\|$. The explicit dependence of (17) on $d$ means that, unlike Theorem 2.2, Theorem 2.12 is not dimension-free. We are unaware of dimension-free concentration results for general sub-$\psi$ processes, or even for sub-gamma processes. The dependence on $d$ instead of $\sqrt{d}$ in the above CSS may seem worrying at first glance, but can be rectified by an appropriate choice of $\lambda_t$. Suppose that $\sup_t\|\Sigma_t\|\leqslant \|\Sigma\|$. For CGF-like $\psi$, choosing

$$\lambda_t \asymp \sqrt{\frac{d\log(1/\epsilon) + \log(1/\alpha)}{\|\Sigma\|t\log(t+1)}},\tag{18}$$

gives a bound that shrinks as $\widetilde{O}(\sqrt{\|\Sigma\|(d + \log(1/\alpha))\log(t)/t})$ as shown by the following lemma. The proof is provided in Appendix C.1.

**Lemma 2.13.** *Let $(X_t)_{t\geqslant 1}$ be $\Sigma_t$-sub-$\psi$ for any CGF-like $\psi$. Let $W_t$ denote the right hand side of (17) and suppose that $\sup_t\|\Sigma_t\|\leqslant \|\Sigma\|$. Then, choosing $\lambda_t$ as in (18) gives $W_t = \widetilde{O}(\sqrt{\|\Sigma\|(d + \log(1/\alpha)\log(t)/t})$.*

As in Sections 2.1 and 2.2, we may use Theorem 2.12 to obtain a bound which achieves optimal iterated-logarithm rates. The details may be found in Appendix B. We focus on sub-gamma random vectors (i.e., $\psi = \psi_{G,c}$). This is mostly without loss of generality, as Howard et al. (2020) demonstrate that any CGF-like $\psi$ obeys $\psi \leqslant a\psi_{G,c}$ for some $a, c > 0$; see Howard et al. (2020, Proposition 1).

**Theorem 2.14.** *Let $(X_t)_{t\geqslant 1}$ be $(\|\Sigma\|, c)$-sub-gamma for some $c \in \mathbb{R}$ and satisfy Assumption 2. Then, for all $\alpha \in (0,1)$, with probability at least $1 - \alpha$, simultaneously for all $t \geqslant 1$,*

$$\left\|\frac{1}{t}\sum_{i\leqslant t}X_i - \mu\right\| \lesssim \sqrt{\frac{\|\Sigma\|(\log(1/\alpha) + \log\log(t) + d)}{t}}.\tag{19}$$

Further, for sub-exponential random vectors (i.e., $\psi = \psi_N$ and $\lambda_{\max} < \infty$) we can obtain the following bound with explicit constants. For $\epsilon = 1/5$, this result is directly comparably to (and tighter than) that of Manole and Ramdas (2023) which, to our knowledge, was previously the tightest known result for sequential concentration of sub-exponential random vectors. Moreover, our result holds under martingale dependence and allows for anisotropy, which theirs does not. The details are again in Appendix B.

**Theorem 2.15.** *Let $(X_t)_{t\geqslant 1}$ be $(\|\Sigma\|, b)$-sub-exponential and satisfy Assumption 2. Fix $0 < \epsilon < 1$. Then, for all $\alpha \in (0,1)$, with probability at least $1 - \alpha$, simultaneously for all $t \geqslant 2b\|\Sigma\|^{-1/2}\sqrt{d\log(1/\epsilon) + \log(1.65/\alpha) + \log(\log_2(t) + 1)}$,*

$$\left\|\frac{1}{t}\sum_{i\leqslant t}X_i - \mu\right\| \leqslant \frac{1.71}{1-\epsilon}\sqrt{\frac{\|\Sigma\|(\log(1.65/\alpha) + \log(\log_2(t) + 1) + d\log(1/\epsilon))}{t}}.\tag{20}$$

Let us compare Theorem 2.15 to Theorem 2.9 for isotropic sub-exponential random vectors, i.e., $\Sigma = \sigma^2 I_d$. Let $r_\alpha(t) = \log(1/\alpha) + \log(\log(t))$. The numerator of the CSS in Theorem 2.15 scales as $\sigma\sqrt{r_\alpha(t) + d}$ and that in Theorem 2.9 scales as $\sigma\sqrt{d\sqrt{r_\alpha(t)} + r_\alpha(t)\sqrt{d}}$, a worse rate. Of course, Theorem 2.9 holds for more general distributions. Fixed-time optimization of Theorem 2.15 can be found in Appendix A.3.

## 2.4 Time-varying means under sub-Gaussianity

Here we drop Assumption 2 and instead allow $\mu_t \equiv \mathbb{E}[X_t|\mathcal{F}_{t-1}]$ to change with time. We are interested in understanding the behavior of $\left\|\sum_{i\leqslant t}(X_i - \mu_i)\right\|$. Our bounds thus far have resulted from applying Proposition 1.1 to families of supermartingales involving predictable free parameters $\lambda_t$ that we may optimize as a function of $t$. This approach does not work when the means are changing, roughly because it doesn't allow us to isolate $\sum_{i\leqslant t}(X_i - \mu_i)$. Instead, we use an approach known as the "method of mixtures," which involves defining a supermartingale $(M_t(\theta, \lambda)_t$ for each $\lambda$ in some set $\Lambda \subset \mathbb{R}$, and then integrating over a well-chosen distribution $\pi$: $M_t(\theta) := \int_{\lambda\in\Lambda}M_t(\theta, \lambda)\pi(d\lambda)$. Fubini's theorem implies that the resulting process $(M_t(\theta))_t$ is again a nonnegative supermartingale.

The method of mixtures is one of the best known techniques for producing time-uniform bounds, dating back at least to Darling and Robbins (1968). In particular, using a two-sided Gaussian mixture, which we do here, was pioneered by Robbins (1970). It has also been used in modern work (cf. Kaufmann and Koolen, 2021; Howard et al., 2021; Waudby-Smith et al., 2024). Of course, not all mixtures will result in a tractable process $(M_t(\theta))_t$ (analytically or computationally). This is doubly true in our case, because our arguments rely on isolating the term $\langle \vartheta, X_i - \mu \rangle$. Therefore, even implicit bounds that might be approximated computationally in the scalar setting do not serve us well. (See for instance Howard et al. (2021) who study using one- and two-sided Gaussian mixtures, Gamma mixtures, one and two-sided beta binomial mixtures, and others.)

When $(X_t)$ are conditionally sub-Gaussian, we can obtain a closed-form bound on $\left\| \sum_{i \leqslant t} (X_i - \mu_i) \right\|$ using a two-sided Gaussian mixture distribution as $\pi$. For technical reasons elucidated in the proof, we can only consider random vectors that are $\sigma_t$-sub-Gaussian (meaning that (4) holds with $\Sigma_t = \sigma_t^2 I_d$). This leads to a bound with width depending on $\sqrt{d}$ instead of $\Sigma_t$.

**Theorem 2.16.** *Let $(X_t)_{t \geqslant 1}$ be conditionally $\sigma_t$-sub-Gaussian where $X_t$ has conditional mean $\mu_t$. Set $H_t = \sum_{i \leqslant t} \sigma_i^2$. Then, for any $a > 0$, $\epsilon \in (0,1)$, and $\alpha \in (0,1)$, we have that with probability $1 - \alpha$, simultaneously for all $t \geqslant 1$,*

$$\left\| \frac{1}{t} \sum_{i \leqslant t} (X_i - \mu_i) \right\| \leqslant \frac{1}{1-\epsilon} \sqrt{\frac{2(1+a^2 H_t)}{a^2 t^2} \left( d \log(1/\epsilon) + \log\left( \sqrt{1+a^2 H_t}/\alpha \right) \right)}. \tag{21}$$

The proof is provided in Appendix C.1. Like Theorem 2.12, we must rely on uniform distributions in the PAC-Bayes argument (note the prior and posterior used in Proposition 1.1 is distinct from the mixing distribution used in the method of mixtures), which accounts the dimension-dependence of the bound. One should compare our bound to Robbins' original mixture (see Robbins, 1970, Equation (17), or Waudby-Smith et al., 2024, Equation (8) for a version closer in spirit and notation to our work here).

### 2.5 Obtaining confidence ellipsoids

Let us end this section by briefly discussing how the techniques developed thus far can be used to obtain bounds in the Mahalanobis norm instead of the $\ell_2$ norm. We use sub-Gaussian random vectors as our primary example. If $(X_t)_{t \geqslant 1}$ are $\Sigma$-sub-Gaussian, then the random vectors $\Sigma^{-1/2} X_t$ are 1-sub-Gaussian. That is, $\mathbb{E}[\exp\{\lambda \langle \theta, \Sigma^{-1/2}(X_t - \mu) \rangle - \psi_N(\lambda) \langle \theta, \theta \rangle | \mathcal{F}_{t-1}] \leqslant 1$ for all $\lambda \in \mathbb{R}$. Theorem 2.3 thus implies the following CSS: with probability $1 - \alpha$, simultaneously for all $t \geqslant 1$,

$$\left\| \frac{1}{t} \sum_{i \leqslant t} X_i - \mu \right\|_{\Sigma^{-1}} = \left\| \frac{1}{t} \sum_{i \leqslant t} \Sigma^{-1/2} X_i - \Sigma^{-1/2} \mu \right\|$$

$$\leqslant 1.21 \sqrt{\frac{d}{t}} + 1.682 \sqrt{\frac{\log(1.65/\alpha) + 2\log(\log_2(t) + 1)}{t}}. \tag{22}$$

Confidence ellipsoids can be desirable because their shape reflects the variance of the distribution. However, the drawback is that in order to be explicitly computed, the variance $\Sigma$ must be both known and fixed. We note that the bound above is distinct in flavor from the self-normalized bounds studied by Abbasi-Yadkori et al. (2011), de la Peña (1999), and Whitehouse et al. (2023). Such bounds aim to achieve self-normalization with respect to an accumulated and possibly random variance process as opposed to a fixed covariance matrix. Bounds similar to (22) can be obtained for other $\Sigma$-sub-$\psi$ conditions, following the techniques in Section 2.3.

## 3 Heavy-tailed random vectors

In this section we suppose the random vectors $(X_t)$ are less well-behaved. We assume only that there exists a finite second moment:

$$\mathbb{E}_P[\|X\|^2 | \mathcal{F}_{t-1}] \leqslant v^2 < \infty, \quad \forall t \geqslant 1. \tag{23}$$

Note that this condition implies that the norm of the mean $\mu$ is bounded by $v$: $\|\mathbb{E}[X|\mathcal{F}_{t-1}]\| \leqslant \mathbb{E}[\|X\| | \mathcal{F}_{t-1}] \leqslant \mathbb{E}[\|X\|^2 | \mathcal{F}_{t-1}]^{1/2} \leqslant v$. Ideally, one would prefer that (23) gets replaced by a bound on the centered moment

$\mathbb{E}_P[\|X - \mu\|^2 | \mathcal{F}_{t-1}]$. Unfortunately, our techniques in this section do not allow for such a constraint so, like Catoni and Giulini (2018) we satisfy ourselves with (23).

In this section we present three main results: two semi-empirical CSSs (Theorems 3.2 and 3.5) and a sequential version of the Catoni-Giulini estimator (Theorem 3.7). All of these CSSs are dimension-free. Throughout this section, we let $\rho_\vartheta$ be a Gaussian with mean $\vartheta$ and variance $\beta^{-1} I_d$. Set $\Sigma_t = \text{Cov}(X_t | \mathcal{F}_{t-1})$.

### 3.1 A first semi-empirical CSS

Our first result is based on the following supermartingale derived using an inequality furnished by Delyon (2009, Proposition 12); see also Howard et al. (2020, Lemma 3). Variants of the resulting process have been studied by Wang and Ramdas (2023, Lemma 5) in the context of CSs, and both Haddouche and Guedj (2023, Lemma 1.3) and Chugg et al. (2023, Corollary 17) in the context of PAC-Bayesian bounds.

**Lemma 3.1.** *Suppose* $(X_t)_{t \geqslant 1} \sim P$ *where* $P$ *is any distribution obeying* (23) *and Assumption 2. Let* $X_i(\theta) = \langle \theta, X_i - \mu \rangle$. *For each fixed* $\theta \in \mathbb{R}^d$, *the process* $S(\theta) = (S_t(\theta))_{t \geqslant 1}$ *is a nonnegative* $P$-*supermartingale, where* $S_t(\theta) = \prod_{i \leqslant t} \exp \{\lambda_i X_i(\theta) - \frac{\lambda_i^2}{6} [X_i^2(\theta) + 2\langle \theta, \Sigma_i \theta \rangle]\}$.

Mixing over $\Theta = \mathbb{R}^d$ with prior $\rho_0$ and posteriors $\rho_\vartheta$, $\vartheta \in \mathbb{S}^{d-1}$, we obtain the following theorem, which is proved in Appendix C.2 (along with Lemma 3.1).

**Theorem 3.2.** *Let* $(X_t)_{t \geqslant 1} \sim P$ *for any* $P$ *obeying* (23) *and Assumption 2. Let* $(\lambda_t)$ *be a predictable sequence in* $(0, \infty)$. *For any* $\beta > 0$, *vector* $x$, *and matrix* $A$, *set*

$$s_\beta(x) = \left(1 + \frac{1}{\beta}\right)(\|x\| + v)^2, \quad \text{and} \quad w_\beta(A) = \|A\| + \frac{\text{Tr}(A)}{\beta}.$$

$s_\beta(x) = (1 + \frac{1}{\beta})(\|x\| + v)^2$ *and* $w_\beta(A) = \|A\| + \text{Tr}(A)/\beta$. *Then, for any* $\alpha \in (0, 1)$, *with probability* $1 - \alpha$, *for all* $t \geqslant 1$,

$$\left\| \frac{\sum_{i \leqslant t} \lambda_i X_i}{\sum_{i \leqslant t} \lambda_i} - \mu \right\| \leqslant \frac{\sum_{i \leqslant t} \lambda_i^2 \{s_\beta(X_i) + 2w_\beta(\Sigma_i)\}}{6 \sum_{i \leqslant t} \lambda_i} + \frac{\beta/2 + \log(1/\alpha)}{\sum_{i \leqslant t} \lambda_i}.$$

Theorem 3.2 is *semi*-empirical in the sense that it depends both on the values $\|X_t\|$ but also on $\|\Sigma_t\|$ and $\text{Tr}(\Sigma_t)$. To explicitly compute the bound, therefore, would require knowledge of these values. Still, a known bound on the variance is required for nonasymptotic concentration of unbounded random variables (even in the scalar case), so fully empirical bounds are impossible.

We suggest setting $\beta = \sqrt{\sup_t \text{Tr}(\Sigma_t)}$ (which is finite because $v$ is finite) and $\lambda_t = \sqrt{\frac{\log(1/\alpha)}{v^2 t \log t}}$ so that the *expected* width of the CSS in Theorem 3.2 scales as $\widetilde{O}(v\sqrt{\log(t) \log(1/\alpha)/t})$. (Here we've used that $w_\beta(\Sigma_t) = O(\mathbb{E}[s_\beta(X_t)|\mathcal{F}_{t-1}]) = O(v^2)$). Note that this bound on the width only holds in expectation and this appears to be unavoidable: in general, with exactly two moments and no more, the empirical variance term may not concentrate.

**Scalar setting.** It is perhaps worth noting that in the univariate setting, Theorem 3.2 complements Lemma 6 in Wang and Ramdas (2023). They develop a confidence sequence for scalar-valued, heavy-tailed random variables by using a process similar to that described in Lemma 3.1. Instead of upper bounding $\mu$ by the raw second moment, however, they solve a quadratic equation to obtain an *anti*-confidence sequence for $\mu$, and their final CS results from taking the complement. Theorem 3.2, meanwhile, results in a slightly looser but more digestible CS.

**Corollary 3.3.** *Let* $X_1, X_2, \ldots$ *be scalar random variables with second moment* $v^2$ *and conditional mean* $\mu = \mathbb{E}[X_t | \mathcal{F}_{t-1}]$. *Let* $(\lambda_t)$ *be a predictable sequence in* $(0, \infty)$. *Then, for any* $\alpha \in (0, 1)$, *with probability* $1 - \alpha$, *for all* $t \geqslant 1$,

$$\left| \frac{\sum_{i \leqslant t} \lambda_i X_i}{\sum_{i \leqslant t} \lambda_i} - \mu \right| \leqslant \frac{\sum_{i \leqslant t} \lambda_i^2 [(\|X_i\| + v)^2 + 2v^2]}{3 \sum_{i \leqslant t} \lambda_i} + \frac{1/2 + \log(1/\alpha)}{\sum_{i \leqslant t} \lambda_i}.$$

We note that despite Corollary 3.3 being closed-form, it may perform worse than the bound of Wang and Ramdas (2023) in practice. First, as noted, to achieve a closed-form bound we deploy some inequalities which loosens the bound. Second, the right hand side may not concentrate. Indeed, we should not expect the sample mean to have sub-Gaussian tail behavior around the true mean.

Considering the fixed-time setting with $n$ observations and setting $\lambda_i = \lambda := \sqrt{\frac{3(\log(1/\alpha)+1/2)}{2nv^2}}$, Corollary 3.3 gives the following CI for heavy-tailed random variables.

**Corollary 3.4.** *Let $X_1, \ldots, X_n$ be scalar random variables with second moment $v^2$ and conditional mean $\mu = \mathbb{E}[X_t|\mathcal{F}_{t-1}]$. Then, for any $\alpha \in (0,1)$, with probability $1 - \alpha$,*

$$\left| \frac{1}{n} \sum_{i \leqslant n} X_i - \mu \right| \lesssim \sqrt{\frac{\log(1/\alpha)}{n}} \left( \frac{\sum_{i \leqslant n}(\|X_i\| + v)^2}{nv} \right).$$

### 3.2 A semi-empirical CSS under symmetry

Let us present a second semi-empirical bound for heavy-tailed random vectors. We assume that the vectors $(X_t)_{t \geqslant 1}$ are conditionally symmetric around the conditional mean $\mu$, i.e., $X_t - \mu \sim -(X_t - \mu)|\mathcal{F}_{t-1}$, and also have two moments. Using a supermartingale identified in de la Peña (1999, Lemma 6.1), we may obtain a result reminiscent of Theorem 3.2 but without the term $w_\beta(\Sigma_t)$ and with tighter constants. Theorem 3.5 is proved in Appendix C.2.

**Theorem 3.5.** *Let $(X_t)_{t \geqslant 1} \sim P$ for any $P$ obeying the moment bound (23) and Assumption 2. Suppose the vectors are also conditionally symmetric: $X_t - \mu \sim -(X_t - \mu)|\mathcal{F}_{t-1}$ for all $t \geqslant 1$. For any positive predictable sequence $(\lambda_t)$ and any $\alpha \in (0,1)$, with probability $1 - \alpha$, for all $t \geqslant 1$,*

$$\left\| \frac{\sum_{i \leqslant t} \lambda_i X_i}{\sum_{i \leqslant t} \lambda_i} - \mu \right\| \leqslant \frac{\sum_{i \leqslant t} \lambda_i^2(\|X_i\| + v)^2 + 1/2 + \log(1/\alpha)}{\sum_{i \leqslant t} \lambda_i}.$$

Of course, one may obtain results analogous to Corollaries 3.3 and 3.4 for Theorem 3.5. We omit these for brevity. A similar discussion as above on how set $\lambda_t$ applies here.

### 3.3 A sequential Catoni-Giulini estimator

Now we proceed to a sequentially-valid version of the estimator originally proposed by Catoni and Giulini (2018). To motivate it, observe that the theorems given thus far rely on the estimators $\widehat{\mu}_t = \sum_i \lambda_i X_i / \sum_i \lambda_i$. However, it is known that such simple linear combinations of samples such as the (possibly weighted) empirical average have sub-optimal finite-sample performance under heavy tails (Catoni, 2012). Other estimators are therefore required. To begin, for a sequence of positive scalars $(\lambda_t)$, define the threshold function

$$\mathsf{th}_t(x) := \frac{(\lambda_t \|x\|) \wedge 1}{\lambda_t \|x\|} x. \tag{24}$$

Our estimator will be

$$\widehat{\mu}_t = \frac{\sum_{i \leqslant t} \lambda_i \mathsf{th}_i(X_i)}{\sum_{i \leqslant t} \lambda_i}. \tag{25}$$

The analysis proceeds by separating the quantity $\langle \vartheta, \mathsf{th}_t(X_t) - \mu \rangle$ into two terms:

$$\langle \vartheta, \mathsf{th}_t(X_t) - \mu \rangle = \underbrace{\langle \vartheta, \mathsf{th}_t(X_t) - \mu_t^{\mathsf{th}} \rangle}_{(i)} + \underbrace{\langle \vartheta, \mu_t^{\mathsf{th}} - \mu \rangle}_{(ii)}, \tag{26}$$

where $\mu_t^{\mathsf{th}} = \mathbb{E}[\mathsf{th}_t(X)|\mathcal{F}_{t-1}]$. The second term is dealt with by the following technical lemma, which is proved in Appendix C.2. Intuitively, the result follows from the fact that $\mathsf{th}_t(x)$ shrinks $x$ towards the origin by an amount proportional to $\lambda_t$.

**Lemma 3.6.** *Term (ii) of (26) obeys:* $\sup_{\vartheta \in \mathbb{S}^{d-1}} \langle \vartheta, \mu_t^{\mathsf{th}} - \mu \rangle = \|\mu_t^{\mathsf{th}} - \mu\| \leqslant \lambda_t v^2$.

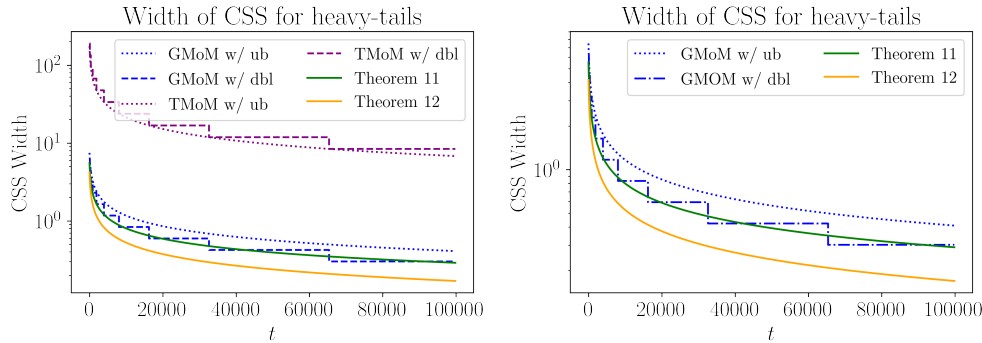

Figure 2: **Left:** Comparison between our sequential Catoni-Giulini estimator (Theorem 3.7), geometric median-of-means (GMoM) (Minsker, 2015), and tournament median-of-means (TMoM) (Lugosi and Mendelson, 2019a). We make the MoM estimators time-uniform in two ways: with a naive union bound (solid lines), and via the doubling method of Duchi and Haque (2024) (DH Doubling). Even though it has optimal (fixed-time) rates, the TMoM estimator suffers because of large constants. **Right:** A closer look at the performance of GMoM estimator compared to Theorem 3.7. We assume that a practitioner knows either $\text{Tr}(\Sigma)$ or $v^2$ (knowing both would imply knowledge of $\|\mu\|$), hence we set $\text{Tr}(\Sigma) = v^2$ in the figures in order to compare their multipliers in the bound. Again, we make the GMoM time-uniform via (i) a union bound, and (ii) Duchi-Haque doubling. Simulation details can be found in Appendix D.

To handle term (i) of (26), we appeal to PAC-Bayesian techniques. Notice that the process $(U(\theta))_{t \geqslant 1}$ defined by $U_t(\theta) = \prod_{i \leqslant t} \exp\{\lambda_i f_i(\theta) - \log \mathbb{E} \exp(\lambda_i f_i(\theta))\}$ where $f_i(\theta) = \langle \theta, \text{th}_i(X_i) - \mu_i^{\text{th}} \rangle$ is a nonnegative $P$-martingale for any $\theta \in \mathbb{R}^d$. Applying Proposition 1.1 with the prior $\rho_0$ and posteriors $\rho_\vartheta$, $\vartheta \in \mathbb{S}^{d-1}$ provides a bound on $\sup_{\vartheta \in \mathbb{S}^{d-1}} \sum_{i \leqslant t} \lambda_i \langle \vartheta, \text{th}_i(X_i) - \mu_i^{\text{th}} \rangle$, which in turn furnishes the following theorem. The details are in Appendix C.2.

**Theorem 3.7.** *Let $(X_t)_{t \geqslant 1}$ obey $\mathbb{E}[\|X_t\|^2 | \mathcal{F}_{t-1}] \leqslant v^2$ and satisfy Assumption 2. Fix $\beta > 0$. Then, for any $\alpha \in (0,1)$, with probability at least $1 - \alpha$, for all $t \geqslant 1$,*

$$\left\| \frac{\sum_{i \leqslant t} \lambda_i \text{th}_i(X_i)}{\sum_{i \leqslant t}^t \lambda_i} - \mu \right\| \leqslant \frac{v^2 \left( 2e^{\frac{2}{\beta}+2} + 1 \right) \sum_{i \leqslant t} \lambda_i^2 + \beta/2 + \log(1/\alpha)}{\sum_{i \leqslant t} \lambda_i}. \tag{27}$$

We find $\beta = 4$ to be a reasonable choice in practice. Consider setting $\lambda_t \asymp \sqrt{\log(1/\alpha)/v^2 t \log t}$. As discussed in Section 2, $\sum_{i \leqslant t} 1/i \log i \asymp \log \log(t)$ and $\sum_{i \leqslant t} 1/\sqrt{i \log i} \asymp \sqrt{t/\log(t)}$, so the width of (27) scales as $\widetilde{O}(\sqrt{v^2 \log(1/\alpha) \log t/t})$. In the fixed time setting, ideal estimators are sub-Gaussian (Lugosi and Mendelson, 2019a, Equation (3.1)), in the sense that they have rate $O(\sqrt{\|\Sigma\| \log(1/\alpha)/n} + \sqrt{\text{Tr}(\Sigma)/n})$ (where we assume that $\Sigma$ is constant over time). For us, setting $\lambda = \lambda_i \propto \sqrt{\log(1/\alpha)/v^2 n}$ for all $i$ yields a slightly larger rate of $O(\sqrt{v^2 \log(1/\alpha)/n})$. This is no surprise: it is well-known that the Catoni-Giulini estimator is not sub-Gaussian, hence the development of the tournament median-of-means (TMoM) estimator (Lugosi and Mendelson, 2019b).

We also provide a stitched version of Theorem 3.7 which achieves iterated logarithm rates. The details are in Appendix B.5.

**Theorem 3.8.** *Let $(X_t)_{t \geqslant 1}$ obey $\mathbb{E}[\|X_t\|^2 | \mathcal{F}_{t-1}] \leqslant v^2$ and satisfy Assumption 2. Let $(\beta_m)_{m \geqslant 1}$ be a sequence of positive scalars such that $\beta_m$ is $\mathcal{F}_{\lfloor \log_2(m) \rfloor}$-predictable. Then, with probability $1 - \alpha$, simultaneously for all $t \geqslant 1$,*

$$\left\| \frac{\sum_{i \leqslant t} \lambda_i \text{th}_i(X_i)}{\sum_{i \leqslant t}^t \lambda_i} - \mu \right\| \leqslant 1.69v \sqrt{\frac{A_{\lfloor \log_2(t) \rfloor}(\beta_{\lfloor \log_2(t) \rfloor}/2 + \log(1.65/\alpha) + \log(\log_2(t) + 1))}{t}}, \tag{28}$$

*where $A_m = 2 \exp(2/\beta_m + 2) + 1$.*

Here, $(\beta_m)$ is a sequence of optimizable constants, though only at geometrically spaced time steps, not every time step. Root-finding may be used to find the optimal value of $\beta_m$ at some value of $t$, though we find that in practice this does not improve the bound much over setting $\beta_m = 4$ for all $m$.

We compare various estimators in Figure 2. The TMoM estimator suffers as a result of large constants, washing out its asymptotic benefits for reasonable sample sizes. Similarly to the experiments in Section 2.1, the TMoM estimator was made time-uniform in two ways: via a naive union bound, and using the doubling technique of Duchi and Haque (2024). Theorem 3.7 outperforms both. We also compare Theorem 3.7 to the *geometric* median-of-means (GMoM) estimator (Minsker, 2015) which achieves a rate of $O(\sqrt{\mathrm{Tr}(\Sigma)\log(1/\alpha)n})$ in the fixed-time setting. We make it time-uniform in the same two ways as TMoM. GMoM has significantly smaller constants than the TMoM estimator, making it much closer to Theorem 3.7 in practice, and sometimes beating it. The stitched Catoni-Giulini estimator, Theorem 3.8, dominates other estimators.

## 4 Summary

We have provided a general framework to derive nonparametric, time-uniform confidence sequences for the mean of a multivariate distribution under martingale dependence using PAC-Bayesian techniques. Our results in light-tailed regimes include dimension-free bounds for sub-Gaussian and log-concave random vectors, bounds for general sub-$\psi$ distributions, and a bound for tracking the time-varying mean of sub-Gaussian distributions. Our results in the heavy-tailed regime include two semi-empirical bounds and a sequentially-valid version of the Catoni-Giulini estimator. We give bounds that achieve optimal iterated-logarithm rates, and also bounds that are optimized for particular sample sizes.

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

## A  Additional results

### A.1  Sub-Gaussian distributions

**Isotropic case.**  Let us instantiate Theorem 2.2 in the isotropic case, meaning that $\Sigma_t = \sigma_t^2 I_d$ for some scalar $\sigma_t$. Then $\|\Sigma_t\| = \sigma_t^2$ and $\mathrm{Tr}(\Sigma_t) = \sigma_t^2 d$. Taking $\beta$ and $\lambda_t$ as in (5), i.e.,

$$\beta = \sqrt{2d\log(1/\alpha)} \ \text{ and } \ \lambda_t = \sqrt{\frac{\beta + 2\log(1/\alpha)}{\sigma^2(1 + d/\beta)t\log(t+1)}},$$

where $\sigma^2 \geqslant \sup_t \sigma_t^2$ gives a width of

$$\widetilde{O}\left(\sqrt{\sigma^2(d + \log(1/\alpha))}\sqrt{\frac{\log t}{t}}\right). \tag{29}$$

We note that the factor of $\sqrt{d}$ is very natural in the isotropic case as, intuitively, the variance is spread evenly in all directions and thus scales with the dimension. See, e.g., Vershynin (2018, Theorem 3.1.1) or Rigollet and Hütter (2023, Theorem 1.19) for the same $\sqrt{d}$ dependence.

### A.2  Log-concave distributions

**Fixed-time optimization.**  Following the approach in Section 2.1, let us consider optimizing our log-concave bound for a fixed time $t = n$. Consider taking

$$\lambda_1 = \cdots = \lambda_n = \sqrt{\frac{2h_\Sigma(\log(2/\alpha))}{nh_\Sigma(1)}},$$

if $n$ is large enough such that $\lambda_1$ is upper bounded by 1. Applying Theorem 2.8 and using the upper bound on $h_\Sigma(r)h_\Sigma(1)$ gives the following result. Again, the bound is tightest at time $t = n$ but remains valid at all other times. We are unaware of previous fixed-time concentration results for log-concave distributions against which we can compare Corollary A.1.

**Corollary A.1.** *Let $(X_t)_{t\geqslant 1}$ satisfy (10) and Assumption 2. Fix any $n$ and $\alpha \in (0,1)$ such that $n \geqslant \sqrt{2h_\Sigma(r_\alpha)/h_\Sigma(1)}$ where $r_\alpha = \log(2/\alpha)$. Then, with probability $1 - \alpha$, simultaneously for all $t \geqslant 1$,*

$$\left\|\frac{\sum_{i\leqslant t}\lambda_i X_i}{\sum_{i\leqslant t}\lambda_i} - \mu\right\| \leqslant 2C\left(\sqrt{\frac{2}{n}} + \sqrt{\frac{2n}{t}}\right)\sqrt{\mathrm{Tr}(\Sigma)\sqrt{r_\alpha} + 3r_\alpha\sqrt{\mathrm{Tr}(\Sigma)\|\Sigma\|}}. \tag{30}$$

### A.3  Sub-$\psi$ distributions

**Fixed-time optimization.**  Again, following the approach in Section 2.1, let us consider $(\Sigma, b)$-sub-exponential random vectors for a constant $\Sigma = \Sigma_t$. Applying Theorem 2.12 at a fixed-time $t = n$ with

$$\lambda = \sqrt{\frac{2d\log(1/\epsilon) + 2\log(1/\alpha)}{\|\Sigma\|n}},$$

gives the following analogue of Corollary 2.5 in the sub-exponential setting.

**Corollary A.2.** *Let $X_1, \ldots, X_n$ be conditionally $(\Sigma, B)$-sub-exponential and satisfy Assumption 2. Fix $\epsilon \in (0,1)$. Then, with probability $1 - \delta$, simultaneously for all $t \geqslant 1$,*

$$\left\|\frac{1}{t}\sum_{i\leqslant t}X_i - \mu\right\| \leqslant \frac{1}{1-\epsilon}\left(\frac{1}{\sqrt{n}} + \frac{\sqrt{n}}{t}\right)\sqrt{\frac{\|\Sigma\|(d\log(1/\epsilon) + \log(1/\alpha))}{2}}. \tag{31}$$

As in Corollary 2.5, this bound is tightest at $t = n$ but remains time-uniform. At $t = n$, taking $\epsilon = 1/2$ we obtain a width of $2\sqrt{2\|\Sigma\|(d\log(2) + \log(1/\alpha)/n}$, which has the optimal dependence on $d$, $n$, and $\|\Sigma\|$ for isotropic sub-exponential random vectors.

**Sub-$\psi$ bounds for non super-Gaussian $\psi$.** Theorem 2.12 gave a CSS for sub-$\psi$ distributions with super-Gaussian $\psi$. While most common $\psi$ functions are super-Gaussian, not all are. For example, consider $\psi_{B,c,d}(\lambda) = \frac{1}{cd}\log\left(\frac{ce^{d\lambda}+de^{-c\lambda}}{c+d}\right)$ for $c, d > 0$ and $\lambda_{\max} = \infty$. This characterizes a sub-Bernoulli distribution (in the sense that $\psi_{B,c,d}$ is the CGF of a centered random variable supported on $-c$ and $d$). It's therefore worth providing a result which holds for general $\psi$ functions, both super-Gaussian and non-super-Gaussian.

The key to deriving Theorem 2.12 was noticing that a super-Gaussian $\psi$ allows us to transform the definition of a sub-$\psi$ distribution (14) from a statement involving vectors in $\mathbb{S}^{d-1}$ to vectors in $\mathbb{B}^d$. This allowed us to use uniform distributions over the ball when applying Proposition 1.1. If $\psi$ is not super-Gaussian then such a transformation isn't possible. We must therefore work with distributions that are defined on the unit sphere. This leads us to the *von Mises-Fisher distribution.* Before we expound on the details of this distribution, let us state the result it enables.

**Theorem A.3.** *Suppose $(X_t)_{t \geqslant 1}$ for $X_t \in \mathbb{R}^d$, $d \geqslant 2$, are sub-$\psi$ and satisfy Assumption 2. Let $(\lambda_t)_{t \geqslant 1}$ be a predictable sequence in $[0, \lambda_{\max})$. Then, with probability at least $1 - \alpha$, for all $t \geqslant 1$,*

$$\left\| \frac{\sum_{i \leqslant t} \lambda_i X_i}{\sum_{i \leqslant t} \lambda_i} - \mu \right\| \leqslant \frac{\sqrt{d}\sum_{i \leqslant t}\psi(\lambda_i)\|\Sigma_i\| + 2\sqrt{d} + \sqrt{d}\log(1/\alpha)}{\frac{2}{3}\sum_{i \leqslant t}\lambda_i}. \tag{32}$$

Notice that unlike Theorem 2.12, the dimension dependence in Theorem A.3 multiplies the width of the bound. That, combined with the prefactor of $3/2$ typically leads to looser bounds than Theorem 2.12, so we encourage practitioners to use the latter when possible.

Let us now introduce the von-Mises Fisher (vMF) distribution (Fisher, 1953) (also known as Langevin distributions, Watamori, 1996). For $x \in \mathbb{S}^{d-1}$, the vMF distribution has density $\gamma(x; \vartheta, \kappa) = C_d(\kappa)\exp(\kappa\langle\vartheta, x\rangle)$, where $C_d(\kappa) = \kappa^{d/2-1}/[(2\pi)^{d/2}I_{d/2-1}(\kappa)]$ is the normalization constant. Here, $I_\ell$ is the modified Bessel function of the first kind of order $\ell$, and $\kappa > 0$ is a scalar "concentration parameter". The expected value of the vMF distribution obeys

$$\mathbb{E}_{\theta \sim \gamma_\vartheta}\theta = \int_{\mathbb{S}^{d-1}} \theta\gamma(\mathrm{d}\theta; \vartheta, \kappa) = A_d(\kappa)\vartheta, \tag{33}$$

where $A_d(\kappa) = I_{d/2}(\kappa)/I_{d/2-1}(\kappa)$. The vMF can be obtained by starting with a multivariate Gaussian and then conditioning on observations with unit norm. We refer to Mardia et al. (2000) for more on the vMF distribution, and for a general introduction to the field of "directional statistics," which considers distributions on the sphere. Moreover, Lemma C.2 in Appendix C proves that

$$D_{\mathrm{KL}}(\gamma(\cdot; \vartheta_1, \kappa)\|\gamma(\cdot; \vartheta_2, \kappa)) \leqslant 2\kappa A_d(\kappa). \tag{34}$$

For an idea of its magnitude, Lemma C.3 proves that $\sqrt{d}A_d(\sqrt{d}) \in (2/3, 1)$. The vMF is only well-defined for $d \geqslant 2$ so we restrict ourselves to this setting. We will often write $\gamma_\vartheta(\cdot) = \gamma(\cdot; \vartheta, \kappa)$, leaving the $\kappa$ implicit.

*Proof of Theorem A.3.* We apply Proposition 1.1 with prior $\gamma(\cdot; 1, \kappa)$ and posteriors $\gamma(\cdot; \vartheta, \kappa)$ for all $\vartheta \in \mathbb{S}^{d-1}$ to the process defined by $M_t(\theta) = \prod_{i \leqslant t}\exp\{\lambda_i\langle\theta, X_i - \mu\rangle - \psi(\lambda_i)\langle\theta, \Sigma_i\theta\rangle\}$. This gives that with probability $1 - \alpha$, for all $t \geqslant 1$,

$$\sum_{i \leqslant t}\lambda_i\int_{\mathbb{S}^{d-1}}\langle\theta, X_i - \mu\rangle\gamma_\vartheta(\mathrm{d}\theta) \leqslant \sum_{i \leqslant t}\psi(\lambda_i)\int_{\mathbb{S}^{d-1}}\langle\vartheta, \Sigma_i\vartheta\rangle\gamma_\vartheta(\mathrm{d}\theta) + 2\kappa A_d(\kappa) + \log(1/\alpha).$$

Using (33) and upper bounding $\langle\vartheta, \Sigma_i\vartheta\rangle$ as $\|\Sigma_i\|$, the above display becomes

$$\sum_{i \leqslant t}A_d(\kappa)\lambda_i\langle\vartheta, X_i - \mu\rangle \leqslant \sum_{i \leqslant t}\psi(\lambda_i)\|\Sigma_i\| + 2\kappa A_d(\kappa) + \log(1/\alpha).$$

Dividing both sides by $A_d(\kappa)\sum_{i \leqslant t}\lambda_i$, taking a supremum over $\vartheta \in \mathbb{S}^{d-1}$, and then taking $\kappa = \sqrt{d}$ and applying Lemma C.3 gives the result. □

# B    Stitched bounds with LIL rate

Here we demonstrate how to achieve CSSs which shrink at a rate of $O(\sqrt{\log(\log(t))/t})$ and proving Theorems 2.3, 2.9, 2.14, 2.15, and 3.8. This is done via "stitching," which is a common technique in the literature on confidence sequences and originates in Howard et al. (2021). Stitching involves applying distinct bounds over geometrically spaced epochs. Bounds in each epoch can be engineered to be tighter than a bound uniform over all time. The bounds are then carefully combined together using a union bound. To proceed, we define our "stitching function" as

$$\ell(m) = (m+1)^2\zeta(2), \quad \text{where} \quad \zeta(2) = \sum_{k=1}^{\infty} k^{-2} \approx 1.645. \tag{35}$$

Note that $\sum_{m\geqslant 0} \frac{1}{\ell(m)} = 1$. In what follows, any function $\ell$ could be used such that $\sum_{m\geqslant 0} \frac{1}{\ell(m)} \leqslant 1$ but here we fix a particular choice for convenience.

## B.1    Stitched Sub-Gamma Bound

Here we consider the sub-$\psi$ bounds of Section 2.1 with $\psi(\lambda) = \psi_{G,c}(\lambda) = \frac{\lambda^2}{2(1-c\lambda)}$, $c \in \mathbb{R}$. We assume that $\Sigma = \Sigma_t$ for all $t$. Apply Theorem 2.12 in each epoch $[2^m, 2^{m+1})$ with a constant parameter $\lambda_m$ and $\alpha_m$. We obtain that with probability $1 - \alpha_m$,

$$\left\| \frac{1}{t}\sum_{i\leqslant t} X_i - \mu \right\| \leqslant \frac{1}{1-\epsilon}\left( \frac{\psi(\lambda_m)\|\Sigma\|t + d\log(1/\epsilon) + r_m}{\lambda_m t} \right) =: g_m(t), \quad \forall t \in [2^m, 2^{m+1}), \tag{36}$$

where $r_m = \log(1/\alpha_m)$. Taking a union bound gives that with probability $1 - \sum_{m=0}^{\infty} \alpha_m$, simultaneously for all $t \geqslant 1$:

$$\left\| \frac{1}{t}\sum_{i\leqslant t} X_i - \mu \right\| \leqslant g_m(t), \quad \text{where} \quad m \leqslant \log_2(t) \leqslant m+1.$$

We take $\alpha_m = \alpha/\ell(m)$ so that $\sum_{m=0}^{\infty} \alpha_m = \alpha$. The remaining work resides in ensuring that $g_m(t)$ shrinks at the desired iterated logarithm rate. We take

$$\lambda_m = \psi^{-1}\left( \frac{d\log(1/\epsilon) + r_m}{2^m\|\Sigma\|} \right),$$

where

$$\psi^{-1}(u) = \frac{2}{c + \sqrt{c^2 + 2/u}}.$$

Then $2r_m/t \geqslant r_m/2^m \geqslant r_m/t$ so

$$\frac{d\log(1/\epsilon) + r_m}{t\|\Sigma\|} \leqslant \psi(\lambda_m) \leqslant 2\frac{d\log(1/\epsilon) + r_m}{t\|\Sigma\|},$$

and since $\psi^{-1}(u)$ is increasing in $u$,

$$\psi^{-1}\left( \frac{d\log(1/\epsilon) + r_m}{2^m\|\Sigma\|} \right) \geqslant \frac{2}{c + \sqrt{c^2 + 2t\|\Sigma\|/(r_m + d\log(1/\epsilon))}}.$$

Therefore,

$$g_m(t) \leqslant \frac{1}{1-\epsilon}\left( \frac{\frac{d\log(1/\epsilon)+r_m}{2^m}t + d\log(1/\epsilon) + r_m}{t\lambda_m} \right)$$

$$\leqslant \frac{3}{1-\epsilon}\left( \frac{d\log(1/\epsilon) + r_m}{t\lambda_m} \right)$$

$$\leqslant \frac{3}{1-\epsilon}\left(c + \sqrt{c^2 + \frac{2t\|\Sigma\|}{d\log(1/\epsilon)+r_m}}\right)\left(\frac{d\log(1/\epsilon)+r_m}{2t}\right).$$

Since $c$ is a constant and is dominated asymptotically by $\frac{t\|\Sigma\|}{d\log(1/\epsilon)+r_m}$, we have (taking $\epsilon = 1/2$ for simplicity),

$$g_m(t) \lesssim \sqrt{\frac{t\|\Sigma\|}{d\log(2)+r_m}}\left(\frac{d\log(2)+r_m}{t}\right) \lesssim \sqrt{\frac{\|\Sigma\|(d+\log(1/\alpha)+\log\log(t))}{t}}.$$

Then notice that

$$r_m = \log(1/\alpha) + \log(\ell(m)) \leqslant \log(1/\alpha) + 2\log(\log_2(t)+1) + \log(1.65),$$

where we've used that $m \leqslant \log_2(t)$. It remains only to check that our choice of $\lambda_m$ is legal, i.e., $\lambda < 1/c$. Using that $\psi^{-1}$ is increasing, we have

$$\lambda_m \leqslant \psi^{-1}\left(\frac{2d\log(2)+2r_m}{t\|\Sigma\|}\right),$$

which is less than $1/c$ iff $1 < \sqrt{1 + t\|\Sigma\|/(d\log(2)+r_m)}$, which holds for all $t \geqslant 1$.

## B.2 Stitched sub-exponential bound

Here we prove Theorem 2.15. Let $\psi(\lambda) = \lambda^2/2$ for all $|\lambda| < 1/b$. Here we make the same choices we did above, but the analysis can be tighter. In this case, $\psi^{-1}(u) = \sqrt{2u}$ so

$$\lambda_m = \sqrt{\frac{d\log(1/\epsilon)+r_m}{2^{m-1}\|\Sigma\|}},$$

and

$$\begin{aligned}
g_m(t) &= \frac{1}{1-\epsilon}\left(\frac{\lambda_m\|\Sigma\|}{2} + \frac{d\log(1/\epsilon)+r_m}{\lambda_m t}\right) \\
&= \frac{\sqrt{\|\Sigma\|(d\log(1/\epsilon)+r_m)}}{1-\epsilon}\left(\frac{1}{2\sqrt{2^{m-1}}} + \frac{\sqrt{2^{m-1}}}{t}\right) \\
&\leqslant \frac{\sqrt{\|\Sigma\|(d\log(1/\epsilon)+r_m)}}{1-\epsilon}\left(\frac{1}{\sqrt{t}} + \frac{1}{\sqrt{2t}}\right) \\
&< \frac{1.71}{1-\epsilon}\sqrt{\frac{\|\Sigma\|(d\log(1/\epsilon)+r_m)}{t}}.
\end{aligned}$$

Finally, recall that we need to ensure that $\lambda_m \leqslant 1/b$, which holds if $t \geqslant 2b\sqrt{\frac{d\log(1/\epsilon)+r_m}{\|\Sigma\|}}$.

## B.3 Stitched sub-Gaussian bound

Let $f_\beta(\Sigma) = \|\Sigma\| + \mathrm{Tr}(\Sigma)/\beta$. Similarly to what was done above, applying Theorem 2.2 in each epoch $[2^m, 2^{m+1})$ with parameters $\lambda_m$, $\alpha_m$, and $\beta_m$ gives

$$\left\|\frac{1}{t}\sum_{i\leqslant t}X_i - \mu\right\| \leqslant \frac{t\psi_N(\lambda_m)f_{\beta_m}(\Sigma) + \beta_m/2 + r_m}{t\lambda_m} =: g_m(t), \quad \forall t \in [2^m, 2^{m+1}), \tag{37}$$

where $r_m = \log(1/\alpha_m)$. As before we set $\alpha_m = \alpha/\ell(m)$ and we take

$$\lambda_m = \frac{\beta_m}{\sqrt{2^m\,\mathrm{Tr}(\Sigma)}}, \quad \text{and} \quad \beta_m = c\sqrt{\frac{r_m\,\mathrm{Tr}(\Sigma)}{\|\Sigma\|}}.$$

Then, noting that $2^m \leqslant t < 2^{m+1}$ so $\lambda_m \leqslant \sqrt{2}\beta_m/\sqrt{t\operatorname{Tr}(\Sigma)}$, we have

$$
\begin{aligned}
g_m(t) &= \frac{\lambda_m f_{\beta_m}(\Sigma)}{2} + \frac{\beta_m/2 + r_m}{t\lambda_m} \\
&\leqslant \frac{\beta_m f_{\beta_m}(\Sigma)}{\sqrt{2t\operatorname{Tr}(\Sigma)}} + \frac{1}{2}\sqrt{\frac{\operatorname{Tr}(\Sigma)}{t}} + \frac{r_m}{\beta_m}\sqrt{\frac{\operatorname{Tr}(\Sigma)}{t}} \\
&= \frac{\beta_m\|\Sigma\|}{\sqrt{2t\operatorname{Tr}(\Sigma)}} + \sqrt{\frac{\operatorname{Tr}(\Sigma)}{2t}} + \frac{1}{2}\sqrt{\frac{\operatorname{Tr}(\Sigma)}{t}} + \frac{r_m}{\beta_m}\sqrt{\frac{\operatorname{Tr}(\Sigma)}{t}} \\
&= c\sqrt{\frac{r_m\|\Sigma\|}{2t}} + \frac{1}{c}\sqrt{\frac{r_m\|\Sigma\|}{t}} + \left(\frac{1}{\sqrt{2}} + \frac{1}{2}\right)\sqrt{\frac{\operatorname{Tr}(\Sigma)}{t}}.
\end{aligned}
$$

Optimizing over $c$ gives $c = 2^{1/4}$ and bounding $r_m$ as above gives the desired result.

### B.4 Stitched log-concave bound

Our strategy is the same as in the previous sections. In this case we have

$$
g_m(t) = \frac{2Ch_\Sigma(1)\lambda_m^2 t + 4Ch_\Sigma(\log(2/\alpha_m))}{\lambda_m t}.
$$

Consider taking

$$
\lambda_m = \kappa\sqrt{\frac{h_\Sigma(\log(2/\alpha_m))}{h_\Sigma(1)2^m}},
$$

for some $\kappa > 0$ to be determined later. Then, using that $2^m \leqslant t$ and $2^m \geqslant 2/t$,

$$
\begin{aligned}
g_m(t) &= 2C\kappa\sqrt{\frac{h_\Sigma(1)h_\Sigma(\log(2/\alpha_m))}{2^m}} + \frac{4C}{\kappa t}\sqrt{h_\Sigma(1)h_\Sigma(\log(2/\alpha_m))2^m} \\
&\leqslant \sqrt{h_\Sigma(1)h_\Sigma(\log(2/\alpha_m))}\left(2C\kappa\sqrt{\frac{2}{t}} + \frac{4C}{\kappa\sqrt{t}}\right).
\end{aligned}
$$

Optimizing over $\kappa$ gives $\kappa = 2^{1/4}$, in which case we obtain

$$
g_m(t) \leqslant 6.73C\sqrt{\frac{h_\Sigma(1)h_\Sigma(\log(2/\alpha_m))}{t}},
$$

where

$$
\begin{aligned}
\log(2/\alpha_m) &\leqslant \log(2\ell(\log_2(t))/\alpha) \\
&\leqslant \log\big(2(\log_2(t)+1)^2 \cdot 1.65/\alpha\big) \\
&= 2\log(\log_2(t)+1) + \log(3.3/\alpha).
\end{aligned}
$$

Recalling that for $u \geqslant 1$, $h_\Sigma(1)h_\Sigma(u) \leqslant \operatorname{Tr}(\Sigma)\sqrt{u} + u\sqrt{\operatorname{Tr}(\Sigma)\|\Sigma\|}$ furnishes the claimed iterated logarithm rate and proves Theorem 2.9.

### B.5 Stitched Catoni-Giulini bound

Let $A_m = 2\exp(2/\beta_m + 2) + 1$. Following the strategy above, here we have

$$
g_m(t) = \frac{v^2 A_m\lambda_m^2 t + \beta_m/2 + r_m}{\lambda_m t} = v^2 A_m\lambda_m + \frac{\beta_m/2 + r_m}{\lambda_m t}.
$$

Take

$$
\lambda_m = c\sqrt{\frac{\beta_m/2 + r_m}{v^2 A_m 2^m}},
$$

for some $c > 0$. Then

$$g_m(t) = v\left(c\sqrt{\frac{A_m(\beta_m/2 + r_m)}{2^m}} + \frac{1}{c}\sqrt{\frac{2^m A_m(\beta_m/2 + r_m)}{t^2}}\right)$$

$$\leqslant v\left(c\sqrt{\frac{2A_m(\beta_m/2 + r_m)}{t}} + \frac{1}{c}\sqrt{\frac{A_m(\beta_m/2 + r_m)}{t}}\right)$$

$$< 1.69v\sqrt{\frac{A_m(\beta_m/2 + r_m)}{t}},$$

if we take $c = 2^{-1/4}$.

## C  Omitted proofs

**Lemma C.1.** *Let $\Sigma \in \mathbb{R}^{d \times d}$ be a covariance matrix. Then*

$$\frac{2\|\Sigma\|}{1 + \sqrt{d}} \leqslant \frac{\text{Tr}(\Sigma^2)}{\text{Tr}(\Sigma)} \leqslant \|\Sigma\|. \tag{38}$$

*Proof.* Let $\Sigma$ have eigenvalues $e_1 \geqslant e_2 \geqslant \ldots \geqslant e_d$. The second inequality is easy: $\text{Tr}(\Sigma^2) = \sum_{1 \leqslant i \leqslant d} e_i^2 \leqslant e_1 \sum_{i \leqslant d} e_i = \|\Sigma\|\text{Tr}(\Sigma)$. As for the first inequality, let $u = \frac{1}{d-1}\sum_{2 \leqslant i \leqslant d} e_i$ be the average of the smallest $d-1$ eigenvalues. Using Jensen's inequality, write

$$\frac{\text{Tr}(\Sigma^2)}{\text{Tr}(\Sigma)} = \frac{\sum_{1 \leqslant i \leqslant d} e_i^2}{\sum_{1 \leqslant i \leqslant d} e_i} \geqslant \frac{e_1^2 + (d-1)u^2}{e_1 + (d-1)u} =: f(u).$$

The minimum of $f$ for $u > 0$ occurs at $u^* = e_1(\sqrt{d}+1)/(d-1)$ giving $f(u^*) = 2e_1/(1+\sqrt{d}) = 2\|\Sigma\|/(1+\sqrt{d})$, which proves the claim. $\qquad\square$

**Lemma C.2.** *The Kullback-Leibler divergence from $\gamma(\vartheta_0, \kappa)$ to $\gamma(\vartheta_1, \kappa)$ satisfies*

$$D_{\text{KL}}(\gamma(\vartheta_1, \kappa)\|\gamma(\vartheta_0, \kappa)) \leqslant 2\kappa A_d(\kappa).$$

*Proof.* Let $X \sim \gamma(\vartheta_1, \kappa)$. By a direct calculation,

$$D_{\text{KL}}(\gamma(\vartheta_1, \kappa)\|\gamma(\vartheta_0, \kappa)) = \mathbb{E}\left[\log \frac{\gamma(X; \vartheta_1, \kappa)}{\gamma(X; \vartheta_0, \kappa)}\right] = \mathbb{E}\left[\kappa\langle\vartheta_1 - \vartheta_0, X\rangle\right] = \kappa\langle\vartheta_1 - \vartheta_0, A_d(\kappa)\vartheta_1\rangle.$$

Since both $\vartheta_0, \vartheta_1$ are on the unit sphere $\mathbb{S}^{d-1}$, the inner product $\langle\vartheta_1 - \vartheta_0, \vartheta_1\rangle$ is upper bounded by 2, which concludes the proof. $\qquad\square$

**Lemma C.3.** *For any $d \geqslant 1$, $\frac{2}{3} < \sqrt{d}A_d(\sqrt{d}) < 1$ where $A_d(\kappa)$ is the vMF constant.*

*Proof.* Note that,

$$\sqrt{d}A_d(\sqrt{d}) = \frac{\sum_{m=0}^{\infty} \sqrt{d}\frac{(\sqrt{d}/2)^{2m+d/2}}{m!\Gamma(m+d/2+1)}}{\sum_{m=0}^{\infty} \frac{(\sqrt{d}/2)^{2m+d/2-1}}{m!\Gamma(m+d/2)}}.$$

Denote the $m^{\text{th}}$ summands of the numerator and denominator by

$$W(m, d) = \sum_{m=0}^{\infty} \sqrt{d}\frac{(\sqrt{d}/2)^{2m+d/2}}{m!\Gamma(m+d/2+1)}, \quad V(m, d) = \frac{(\sqrt{d}/2)^{2m+d/2-1}}{m!\Gamma(m+d/2)}.$$

Then,

$$\frac{W(m, d)}{V(m, d)} = \frac{d/2}{m + d/2} \leqslant 1,$$

equality only when $m = 0$. Therefore,

$$\sqrt{d}A_d(\sqrt{d}) = \frac{\sum_{m=0}^{\infty} V(m,d)\frac{d/2}{m+d/2}}{\sum_{m=0}^{\infty} V(m,d)} < 1.$$

Further, observe that

$$\frac{W(m,d)}{V(m+1,d)} = 2(m+1).$$

Therefore

$$\sqrt{d}A_d(\sqrt{d}) = \frac{\sum_{m=1}^{\infty} 2mV(m,d)}{V(0,d) + \sum_{m=1}^{\infty} V(m,d)},$$

and

$$\frac{1}{\sqrt{d}A_d(\sqrt{d})} = \frac{V(0,d) + \sum_{m=1}^{\infty} V(m,d)}{\sum_{m=1}^{\infty} 2mV(m,d)} < \frac{V(0,d) + \sum_{m=1}^{\infty} V(m,d)}{\sum_{m=1}^{\infty} 2V(m,d)}$$

$$< \frac{V(0,d)}{2V(1,d)} + \frac{1}{2} = 1 + \frac{1}{2} = \frac{3}{2}.$$

Thus we conclude that $\frac{2}{3} < \sqrt{d}A_d(\sqrt{d}) < 1$ for all $d \geqslant 1$. $\qquad\square$

### C.1 Proofs for Section 2

**Proof of Theorem 2.2** By definition of sub-Gaussianity (4), if $X_1, X_2, \ldots \sim P$ are $\Sigma_t$-sub-Gaussian, then the process $(H_t(\theta))_{t \geqslant 1}$ where

$$H_t(\theta) = \prod_{i \leqslant t} \exp\left\{ \lambda_i \langle \theta, X_i - \mu \rangle - \frac{\lambda_i^2}{2} \langle \theta, \Sigma_i \theta \rangle \right\},$$

is a supermartingale for all $\theta \in \mathbb{R}^d$. Let $\rho_\vartheta$ be a Gaussian centered at $\vartheta$ with covariance $\beta^{-1} I_d$. Applying Proposition 1.1 with the prior $\rho_0$ and family of posteriors $\rho_\vartheta$, $\vartheta \in \mathbb{S}^{d-1}$, we obtain that with probability $1 - \alpha$, simultaneously for all $t \geqslant 1$ and $\vartheta \in \mathbb{S}^{d-1}$,

$$\int \sum_{i \leqslant t} \lambda_i \langle \theta, X_i - \mu \rangle \rho_\vartheta(\mathrm{d}\theta) \leqslant \int \sum_{i \leqslant t} \frac{\lambda_i^2}{2} \langle \theta, \Sigma_i \theta \rangle \rho_\vartheta(\mathrm{d}\theta) + D_{\mathrm{KL}}(\rho_\vartheta \| \rho_0) + \log(1/\alpha)$$

$$\leqslant \sum_{i \leqslant t} \frac{\lambda_i^2}{2} (\langle \vartheta, \Sigma_i \vartheta \rangle + \beta^{-1} \mathrm{Tr}(\Sigma_i)) + \frac{\beta}{2} + \log(1/\alpha),$$

where we've used the formula

$$D_{\mathrm{KL}}(N(\vartheta_1, \Sigma_1) \| N(\vartheta_2, \Sigma_2)) = \frac{1}{2}\left( \mathrm{Tr}(\Sigma_2^{-1}\Sigma_1) + \langle \vartheta_2 - \vartheta_1, \Sigma_2^{-1}(\vartheta_2 - \vartheta_1) \rangle - d + \log\frac{|\Sigma_2|}{|\Sigma_1|} \right). \tag{39}$$

The symmetry of the Gaussian distribution implies that $\int \sum_{i \leqslant t} \lambda_i \langle \theta, X_i - \mu \rangle \rho_\vartheta(\mathrm{d}\theta) = \sum_{i \leqslant t} \lambda_i \langle \vartheta, X_i - \mu \rangle$. Noticing that $\langle \vartheta, \Sigma_t \vartheta \rangle \leqslant \|\Sigma_t\|$, we obtain that with probability $1 - \alpha$, simultaneously for all $t \geqslant 1$,

$$\left\| \sum_{i \leqslant t} \lambda_i (X_i - \mu) \right\| = \sup_{\vartheta \in \mathbb{S}^{d-1}} \sum_{i \leqslant t} \lambda_i \langle \vartheta, X_i - \mu \rangle$$

$$\leqslant \sum_{i \leqslant t} \frac{\lambda_i^2}{2} (\|\Sigma_i\| + \beta^{-1} \mathrm{Tr}(\Sigma_i)) + \frac{\beta}{2} + \log\left(\frac{1}{\alpha}\right),$$

which, after rearranging, is the claimed inequality.

**Proof of Theorem 2.8** We take our parameter space in Proposition 1.1 to be $\Theta = \mathbb{R}^d$. Let $\nu$ Gaussian with mean 0 and covariance $\beta^{-1}\Sigma$ and let $\overline{\rho}_u$ be a truncated Gaussian with mean $u \in \Sigma^{1/2}\mathbb{S}^{d-1}$, covariance $\beta^{-1}\Sigma$, and radius $r > 0$. Being slightly loose with notation and writing $\mathrm{d}\overline{\rho}_u$ for the density of $\overline{\rho}_u$, the density of the truncated normal can be written as

$$\mathrm{d}\overline{\rho}_u(x) = \frac{\mathbf{1}\{\|x - u\| \leqslant r\}}{Z}\mathrm{d}\rho_u,$$

where $Z$ is some normalizing constant and $\rho_u$ is the usual non-truncated Gaussian. We follow Zhivotovskiy (2024) in our calculation of the KL-divergence for truncated Gaussians. For a vector $u \in \Sigma^{1/2}\mathbb{S}^{d-1}$, the KL-divergence between a truncated normal and $\nu$ is therefore

$$\begin{aligned}
D_{\mathrm{KL}}(\overline{\rho}_u\|\nu) &= \int \log\left(\frac{1}{Z}\frac{\mathrm{d}\rho_u}{\mathrm{d}\nu}(\theta)\right)\overline{\rho}_u(\mathrm{d}\theta)\\
&= \log\left(\frac{1}{Z}\right) + \frac{1}{2}\int(-\langle\theta - u, \beta\Sigma^{-1}(\theta - u)\rangle + \langle\theta, \beta\Sigma^{-1}\theta\rangle)\overline{\rho}_u(\mathrm{d}\theta)\\
&= \log\left(\frac{1}{Z}\right) + \frac{\beta}{2}\int(2\langle\theta, \Sigma^{-1}u\rangle - \langle u, \Sigma^{-1}u\rangle)\overline{\rho}_u(\mathrm{d}\theta)\\
&= \log\left(\frac{1}{Z}\right) + \frac{\beta\langle u, \Sigma^{-1}u\rangle}{2} = \log\left(\frac{1}{Z}\right) + \frac{\beta}{2},
\end{aligned}$$

where we've used that $u = \Sigma^{1/2}\vartheta$ for some $\vartheta \in \mathbb{S}^{d-1}$ so $\langle u, \Sigma^{-1}u\rangle = \langle\vartheta, \vartheta\rangle = 1$. We also have $Z = \Pr(\|\theta - u\| \leqslant r)$ where $\theta \sim \rho_u$. Equivalently, $Z = \Pr(\|Y\| \leqslant r)$ where $Y$ is a normal with mean 0 and covariance $\beta^{-1}\Sigma$. Hence $1 - Z = \Pr(\|Y\| > r) \leqslant \mathbb{E}\|Y\|^2/r^2 = \beta^{-1}\mathrm{Tr}(\Sigma)/r^2$. Thus, taking $r = \sqrt{2\beta^{-1}\mathrm{Tr}(\Sigma)}$ yields $Z \geqslant 1/2$ and we obtain

$$D_{\mathrm{KL}}(\overline{\rho}_u\|\nu) \leqslant \log(2) + \frac{\beta}{2}. \tag{40}$$

Now, consider the process $(L_t(\theta))$ defined as

$$L_t(\theta) = \prod_{i\leqslant t}\exp\left\{\lambda_i\langle\theta, \Sigma^{-1/2}X_i\rangle - \log\mathbb{E}[\exp\left(\lambda_i\langle\theta, \Sigma^{-1/2}X\rangle\right)|\mathcal{F}_{i-1}]\right\}, \tag{41}$$

which is a nonnegative martingale with initial value 1 as long as

$$\log\mathbb{E}\left[\exp\left(\lambda_t\langle\theta, \Sigma^{-1/2}X\rangle\right)|\mathcal{F}_{t-1}\right] < \infty.$$

$(L_t(\theta)$ is a product of exponentials, each with expected value 1.) We want to use Lemma 2.7 to bound this term. Note that by the log-concavity condition, we have

$$\left\|\langle\theta, \Sigma^{-1/2}X_t\rangle - \mathbb{E}[\langle\theta, \Sigma^{-1/2}X\rangle|\mathcal{F}_{t-1}]\right\|_{\Phi_1} = \left\|\langle\Sigma^{-1/2}\theta, X_t - \mu\rangle\right\|_{\Phi_1} \leqslant C\|\theta\|.$$

Moreover, if $\theta \sim \overline{\rho}_u$ then by definition of the truncated Gaussian and using that $\|u\| = \|\Sigma^{1/2}\vartheta\| \leqslant \sqrt{\|\Sigma\|}$,

$$\|\theta\| \leqslant r + \|u\| \leqslant r + \sqrt{\|\Sigma\|} = \sqrt{2\beta^{-1}\mathrm{Tr}(\Sigma)} + \sqrt{\|\Sigma\|} := L_\beta.$$

Combining this with Lemma 2.7 we obtain

$$\log\mathbb{E}\left[\exp\left(\lambda_t\langle\theta, \Sigma^{-1/2}X\rangle\right)|\mathcal{F}_{t-1}\right] \leqslant \lambda_t\langle\theta, \Sigma^{-1/2}\mu\rangle + 4\lambda_t^2C^2L_\beta^2, \tag{42}$$

if

$$|\lambda_t| \leqslant \frac{1}{2CL_\beta}.$$

(Note that the first term on the right hand side of (42) comes from the mean $\mathbb{E}[\langle\theta,\Sigma^{-1/2}X\rangle|\mathcal{F}_{t-1}]$ that lives on the left hand side in Lemma 2.7.) Now, applying Proposition 1.1 with $L_t(\theta)$ and using (42) gives that with probability $1-\alpha$, for all $u\in\Sigma^{1/2}\mathbb{S}^{d-1}$ and $t\geqslant 1$,

$$\int\sum_{i\leqslant t}\lambda_i\langle\theta,\Sigma^{-1/2}X_i\rangle\overline{\rho}_u(\mathrm{d}\theta)$$

$$\leqslant\int\sum_{i\leqslant t}\log\mathbb{E}\left[\exp\left(\lambda_i\langle\theta,\Sigma^{-1/2}X\rangle\right)|\mathcal{F}_{t-1}\right]\overline{\rho}_u(\mathrm{d}\theta)+\frac{\beta}{2}+\log\left(\frac{2}{\alpha}\right)$$

$$\leqslant\sum_{i\leqslant t}\lambda_i\langle u,\Sigma^{-1/2}\mu\rangle+4C^2L_\beta^2\sum_{i\leqslant t}\lambda_i^2+\frac{\beta}{2}+\log\left(\frac{2}{\alpha}\right),$$

where the final line uses that $\overline{\rho}_u$ is symmetric hence $\int\langle\theta,\Sigma^{-1/2}\mu\rangle\overline{\rho}_u(\mathrm{d}\theta)=\langle u,\Sigma^{-1/2}\mu\rangle$. From here, since $u=\Sigma^{1/2}\vartheta$ for some $\vartheta\in\mathbb{S}^{d-1}$, the above inequality rearranges to read

$$\sum_{i\leqslant t}\lambda_i\langle\vartheta,X_i-\mu\rangle\leqslant 4C^2L_\beta^2\sum_{i\leqslant t}\lambda_i^2+\frac{\beta}{2}+\log\left(\frac{2}{\alpha}\right).$$

Consider setting $\lambda_t=\frac{\widehat{\lambda}_t}{2CL_\beta}$, where $0<\widehat{\lambda}_t\leqslant 1$. Since $\vartheta$ was arbitrary in the above display, we obtain that with probability $1-\alpha$, for all $t\geqslant 1$,

$$\left\|\frac{\sum_{i\leqslant t}\widehat{\lambda}_iX_i}{\sum_{i\leqslant t}\widehat{\lambda}_i}-\mu\right\|\leqslant\frac{2CL_\beta(\sum_{i\leqslant t}\widehat{\lambda}_i^2+\beta/2+\log(2/\alpha))}{\sum_{i\leqslant t}\widehat{\lambda}_i}.$$

Set $\ell=\log(2/\alpha)$ and consider choosing $\beta=2\ell$, in which case $L_\beta\leqslant\sqrt{\mathrm{Tr}(\Sigma)}+\sqrt{\|\Sigma\|}=h_\Sigma(1)$ and the above display becomes

$$\left\|\frac{\sum_{i\leqslant t}\widehat{\lambda}_iX_i}{\sum_{i\leqslant t}\widehat{\lambda}_i}-\mu\right\|\leqslant\frac{2C(\sqrt{\mathrm{Tr}(\Sigma)}+\sqrt{\|\Sigma\|})\sum_{i\leqslant t}\widehat{\lambda}_i^2}{\sum_{i\leqslant t}\widehat{\lambda}_i}+\frac{4C(\sqrt{\mathrm{Tr}(\Sigma)\ell}+\ell\sqrt{\|\Sigma\|})}{\sum_{i\leqslant t}\widehat{\lambda}_i}$$

$$=\frac{2Ch_\Sigma(1)\sum_{i\leqslant t}\widehat{\lambda}_i^2+4Ch_\Sigma(u)}{\sum_{i\leqslant t}\widehat{\lambda}_i},$$

which is the desired bound.

**Proof of Theorem 2.12** By Lemma 2.10 we may apply Proposition 1.1 with the process defined by

$$M_t(\theta)=\prod_{i\leqslant t}\exp\{\lambda_i\langle\theta,X_i-\mu\rangle-\psi(\lambda_i)\langle\theta,\Sigma_i\theta\rangle\},$$

for all $\theta\in\mathbb{B}^d$ (not just for all $\theta\in\mathbb{S}^{d-1}$ as is suggested by (14)). Let $\rho_\vartheta$ be a uniform distribution centered at $\vartheta\in(1-\epsilon)\mathbb{S}^{d-1}\subset\mathbb{B}^d$ with radius $\epsilon$. Proposition 1.1 gives that with probability $1-\alpha$, for all $\vartheta\in(1-\epsilon)\mathbb{S}^{d-1}$,

$$\sum_{i\leqslant t}\lambda_i\int\langle\theta,X_i-\mu\rangle\rho_\vartheta(\mathrm{d}\theta)\leqslant\sum_{i\leqslant t}\psi(\lambda_i)\int\langle\theta,\Sigma_i\theta\rangle\rho_\vartheta(\mathrm{d}\theta)+d\log\left(\frac{1}{\epsilon}\right)+\log\left(\frac{1}{\alpha}\right),$$

where we've used the KL-divergence as calculated in (15). If $\theta\sim\rho_\vartheta$ then $\|\theta\|\leqslant\|\vartheta\|+\epsilon\leqslant 1$ by definition of $\rho_\vartheta$ and $\langle\theta,\Sigma_i\theta\rangle\leqslant\sup_{\theta,\vartheta\in\mathbb{S}^{d-1}}\langle\theta,\Sigma_i\vartheta\rangle\leqslant\|\Sigma_i\|$. Moreover, since $\rho_\vartheta$ is symmetric,

$$\sup_{\vartheta\in(1-\epsilon)\mathbb{S}^{d-1}}\sum_{i\leqslant t}\lambda_i\int\langle\theta,X_i-\mu\rangle\rho_\vartheta(\mathrm{d}\theta)$$

$$=\sup_{\vartheta\in(1-\epsilon)\mathbb{S}^{d-1}}\left\langle\vartheta,\sum_{i\leqslant t}\lambda_i(X_i-\mu)\right\rangle$$

$$= (1 - \epsilon) \left\| \sum_{i \leqslant t} \lambda_i (X_i - \mu) \right\|.$$

Therefore, with probability $1 - \alpha$, for all $t \geqslant 1$,

$$\left\| \frac{\sum_i \lambda_i X_i}{\sum_i \lambda_i} - \mu \right\| \leqslant \frac{\sum_{i \leqslant t} \psi(\lambda_i) \|\Sigma_i\| + d \log(1/\epsilon) + \log(1/\alpha)}{(1 - \epsilon) \sum_i \lambda_i},$$

which is the desired result.

**Proof of Lemma 2.13** If $\psi$ is CGF-like, then Howard et al. (2020, Proposition 1) shows that there exists some $a, c > 0$ such that

$$\psi(\lambda) \leqslant a\psi_{G,c}(\lambda) = \frac{a\lambda^2}{2(1 - c\lambda)}.$$

If $\lambda_t \xrightarrow{t \to \infty} 0$, then

$$\psi(\lambda_t)/\psi_N(\lambda_t) \leqslant \frac{a}{1 - c\lambda_t} \xrightarrow{t \to \infty} a.$$

Therefore, we may write $\psi(\lambda_t)/\psi_N(\lambda_t) = a + u_t$ for some $u_t$ that goes to 0 as $t \to \infty$. Therefore,

$$\begin{aligned}
\sum_{i \leqslant t} \psi(\lambda_i) \|\Sigma_i\| &= \sum_{i \leqslant t} \frac{\psi(\lambda_i)}{\psi_N(\lambda_i)} \psi_N(\lambda_i) \|\Sigma_i\| \\
&= \sum_{i \leqslant t} \psi_N(\lambda_i) \|\Sigma_i\| (a + u_i) \\
&\lesssim (d + r) \sum_{i \leqslant t} \frac{a + u_i}{i \log(i + 1)} \\
&\lesssim (d + r) \log \log(t).
\end{aligned}$$

Hence,

$$\begin{aligned}
W_t &= \frac{\sum_{i \leqslant t} \psi(\lambda_i) \|\Sigma_i\| + d \log(1/\epsilon) + \log(1/\alpha)}{(1 - \epsilon) \sum_i \lambda_i} \\
&\lesssim \frac{(d + r) \log \log(t) + d + r}{\sqrt{\|\Sigma\|(d \log(1/\epsilon) + r) t / \log(t) \|\Sigma\|}} \\
&= \widetilde{O} \left( \sqrt{\frac{\|\Sigma\|(d + r) \log t}{t}} \right),
\end{aligned}$$

which is the desired rate.

**Proof of Theorem 2.16** Let $X_t$ be conditionally $\sigma_t$-sub-Gaussian. That is,

$$\sup_{v \in \mathbb{S}^{d-1}} \mathbb{E}_P[\exp(\lambda \langle v, X_t - \mu_t \rangle) | \mathcal{F}_{t-1}] \leqslant \exp\left(\frac{\lambda^2 \sigma_t^2}{2}\right). \tag{43}$$

Note that we allow the sub-Gaussian parameter to change at each timestep. From (43) it follows that the process defined by

$$M_t(\theta, \lambda) = \prod_{i \leqslant t} \exp\left\{ \lambda \langle \theta, X_i - \mu_i \rangle - \frac{\lambda^2 \sigma_i^2}{2} \right\},$$

is a nonnegative $P$-supermartingale for all $\lambda \in \mathbb{R}$. We will consider the supermartingale resulting from mixing over a Gaussian:

$$M_t(\theta) := \int_{\lambda \in \mathbb{R}} M_t(\theta, \lambda) \pi(\lambda; 0, a^2) \mathrm{d}\lambda, \tag{44}$$

where $\pi$ is the density of a univariate Gaussian with mean 0 variance $a^2$. To compute $M_t(\theta)$ let $D_t(\theta) = \sum_{i \leqslant t} \langle \theta, X_i - \mu_i \rangle$, $H_t = \sum_{i \leqslant t} \sigma_i^2$, and write

$$
\begin{aligned}
M_t(\theta) &= \frac{1}{a\sqrt{2\pi}} \int_{\lambda \in \mathbb{R}} \exp\left\{ \lambda D_t(\theta) - \frac{\lambda^2}{2} H_t \right\} \exp\left\{ -\frac{\lambda^2}{2a^2} \right\} \mathrm{d}\lambda \\
&= \frac{1}{a\sqrt{2\pi}} \int_{\lambda \in \mathbb{R}} \exp\left\{ \frac{2\lambda a^2 D_t(\theta) - \lambda^2(1 + H_t a^2)}{2a^2} \right\} \mathrm{d}\lambda.
\end{aligned}
$$

Put $u_t = 1 + H_t a^2$ and $v_t = a^2 D_t(\theta)$ and note that these are constants with respect to $\lambda$. Rewrite the above as

$$
\begin{aligned}
M_t(\theta) &= \frac{1}{a\sqrt{2\pi}} \int_{\lambda \in \mathbb{R}} \exp\left\{ \frac{-\lambda^2 u_t + 2\lambda v_t}{2a^2} \right\} \mathrm{d}\lambda \\
&= \frac{1}{a\sqrt{2\pi}} \int_{\lambda \in \mathbb{R}} \exp\left\{ \frac{-(\lambda^2 - 2\lambda v_t/u_t)}{2a^2/u_t} \right\} \mathrm{d}\lambda \\
&= \frac{1}{a\sqrt{2\pi}} \int_{\lambda \in \mathbb{R}} \exp\left\{ \frac{-(\lambda - v_t/u_t)^2 + (v_t/u_t)^2}{2a^2/u_t} \right\} \mathrm{d}\lambda \\
&= \frac{1}{a\sqrt{2\pi}} \int_{\lambda \in \mathbb{R}} \exp\left\{ \frac{-(\lambda - v_t/u_t)^2}{2a^2/u_t} \right\} \mathrm{d}\lambda \exp\left\{ \frac{v_t}{2a^2 u_t} \right\} \\
&= \frac{1}{\sqrt{u_t}} \exp\left\{ \frac{v_t^2}{2a^2 u_t} \right\} \\
&= \exp\left\{ \frac{v_t^2}{2a^2 u_t} - \frac{1}{2} \log(u_t) \right\},
\end{aligned}
$$

where the penultimate equality follows because the integrand is proportional to the density of a Gaussian with mean $v_t/u_t$ and variance $a^2/u_t$. We conclude that

$$
M_t(\theta) = \exp\left\{ \frac{a^2 D_t(\theta)}{2(1 + a^2 H_t)} - \log\sqrt{1 + a^2 H_t} \right\}, \tag{45}
$$

is a nonnegative $P$-supermartingale. We can now apply Proposition 1.1 with uniform distributions as we did for sub-$\psi$ distributions in Section 2.3. Let $\rho_\vartheta$ be a uniform distribution over the unit sphere of radius $\epsilon$ centered at $\vartheta \in (1 - \epsilon)\mathbb{S}^{d-1}$. Using (15) to bound the KL-divergence, we obtain that for all $\vartheta \in \mathbb{S}^{d-1}$, with probability $1 - \alpha$, simultaneously for all $t \geqslant 1$,

$$
\begin{aligned}
\int_{\mathbb{S}^{d-1}} \frac{a^2 D_t^2(\theta)}{2(1 + a^2 H_t)} \rho_\vartheta(\mathrm{d}\theta) &\leqslant \int_{\mathbb{S}^{d-1}} \log\sqrt{1 + a^2 H_t} \rho_\vartheta(\mathrm{d}\theta) + d\log(1/\epsilon) + \log(1/\alpha) \\
&= \log\sqrt{1 + a^2 H_t} + d\log(1/\epsilon) + \log(1/\alpha).
\end{aligned} \tag{46}
$$

We work with (43) instead of the more general definition in (4) because we do not want the right hand side to depend on $\theta$. If it did then $H_t$ would be a function of $\theta$, and the integral on the left hand side of (46) would become too complex to solve in closed-form. Continuing with the proof, since $H_t$ is not a function of $\theta$, the above display rearranges to read

$$
\int_{\mathbb{S}^{d-1}} D_t^2(\theta) \rho_\vartheta(\mathrm{d}\theta) \leqslant \frac{2(1 + a^2 H_t)}{a^2} \left( d\log(1/\epsilon) + \log\left( \sqrt{1 + a^2 H_t}/\alpha \right) \right).
$$

Taking square roots of both sides, Jensen's inequality implies that

$$
\int_{\mathbb{S}^{d-1}} D_t(\theta) \gamma_\vartheta(\mathrm{d}\theta) \leqslant \left( \frac{2(1 + a^2 H_t)}{a^2} \left( d\log(1/\epsilon) + \log\left( \sqrt{1 + a^2 H_t}/\alpha \right) \right) \right)^{1/2}.
$$

Recalling the definition of $D_t(\theta)$ and integrating the left hand side as in the proof of Theorem 2.12 we obtain that with probability $1 - \alpha$, simultaneously for all $t \geqslant 1$:

$$
\left\| \sum_{i \leqslant t} (X_i - \mu_i) \right\| \leqslant \frac{1}{1 - \epsilon} \left( \frac{2(1 + a^2 H_t)}{a^2} \left( d\log(1/\epsilon) + \log\left( \sqrt{1 + a^2 H_t}/\alpha \right) \right) \right)^{1/2}.
$$

Dividing both sides by $t$ completes the proof.

### C.2 Proofs for Section 3

**Proof of Lemma 3.1** Delyon (2009, Equation (52)) demonstrates that for all $x \in \mathbb{R}$, $\exp\left(x - x^2/6\right) \leqslant 1 + x + x^2/3$. If we take $x = \lambda_i X_i(\theta)$, then applying expectations (w.r.t. $P$) yields

$$
\mathbb{E}_P\left[\exp\left\{\lambda_i X_i(\theta) - \frac{\lambda_i^2}{6} X_i^2(\theta)\right\} \bigg| \mathcal{F}_{i-1}\right]
$$
$$
\leqslant 1 + \lambda_i \mathbb{E}_P[X_i(\theta)|\mathcal{F}_{i-1}] + \frac{\lambda_i^2}{3}\mathbb{E}_P[X_i^2(\theta)|\mathcal{F}_{i-1}]
$$
$$
= 1 + \frac{\lambda_i^2}{3}\mathbb{E}_P[X_i^2(\theta)|\mathcal{F}_{i-1}] \leqslant \exp\left\{\frac{\lambda_i^2}{3}\mathbb{E}_P[X_i^2(\theta)|\mathcal{F}_{i-1}]\right\}.
$$

From here, note that

$$
\mathbb{E}_P[X_i^2(\theta)|\mathcal{F}_{i-1}] = \mathbb{E}_P[\theta^\top(X_i - \mu)(X_i - \mu)^\top \theta|\mathcal{F}_{i-1}]
$$
$$
= \theta^\top \mathbb{E}_P[(X_i - \mu)(X_i - \mu)^\top|\mathcal{F}_{i-1}]\theta = \theta^\top \Sigma\theta.
$$

Applying this to the display above and rearranging yields that

$$
\mathbb{E}_P \exp\left\{\lambda_i X_i(\theta) - \frac{\lambda_i^2}{6}X_i^2(\theta) - \frac{\lambda_i^2}{3}\langle\theta, \Sigma\theta\rangle \bigg| \mathcal{F}_{i-1}\right\} \leqslant 1,
$$

which in turn implies that $S(\theta)$ is a supermartingale.

**Proof of Theorem 3.2** Let $\rho_\vartheta$ be a Gaussian with mean $\vartheta$ and covariance $\beta^{-1}I_d$. Apply Proposition 1.1 to the supermartingale in Lemma 3.1. We obtain that with probability $1 - \alpha$, for all $t \geqslant 1$ and $\vartheta \in \mathbb{S}^{d-1}$,

$$
\sum_{i \leqslant t} \lambda_i \int X_i(\theta)\rho_\vartheta(\mathrm{d}\theta) \leqslant \sum_{i \leqslant t} \frac{\lambda_i^2}{6}\int X_i^2(\theta) + 2\langle\theta, \Sigma_i\theta\rangle\rho_\vartheta(\mathrm{d}\theta) + \beta/2 + \log(1/\alpha).
$$

Now, let $M_i$ be the matrix $(X_i - \mu)(X_i - \mu)^\top$ and write

$$
\int X_i^2(\theta)\rho_\vartheta(\mathrm{d}\theta) = \int \theta^\top M_i\theta\rho_\vartheta(\mathrm{d}\theta)
$$
$$
= \vartheta^\top M_i\vartheta + \beta^{-1}\operatorname{Tr}(M_i)
$$
$$
\leqslant \|M_i\| + \beta^{-1}\operatorname{Tr}(M_i)
$$
$$
= \|X_i - \mu\|^2(1 + \beta^{-1})
$$
$$
\leqslant (\|X_i\| + v)^2(1 + \beta^{-1}).
$$

Moreover,

$$
\int \langle\theta, \Sigma_i\theta\rangle\rho_\vartheta(\mathrm{d}\theta) = \vartheta^\top \Sigma_i\vartheta + \beta^{-1}\operatorname{Tr}(\Sigma_i) \leqslant \|\Sigma_i\| + \beta^{-1}\operatorname{Tr}(\Sigma_i).
$$

Putting this all together, we obtain that with probability $1 - \alpha$, for all $\vartheta \in \mathbb{S}^{d-1}$ and all $t \geqslant 1$,

$$
\sum_{i \leqslant t} \lambda_i X_i^2(\vartheta) \leqslant (1 + \beta^{-1})\sum_{i \leqslant t} \frac{\lambda_i^2}{6}(\|X_i\| + v)^2 + \sum_{i \leqslant t} \frac{\lambda_i^2}{3}(\|\Sigma_i\| + \beta^{-1}\operatorname{Tr}(\Sigma_i)) + \frac{\beta}{2} + \log\left(\frac{1}{\alpha}\right).
$$

Noticing that supremum over all $\theta \in \mathbb{S}^{d-1}$ of the left hand side equals $\left\|\sum_{i \leqslant t} \lambda_i(X_i - \mu)\right\|$ and then dividing through by $\sum_{i \leqslant t} \lambda_i$ gives the result.

**Proof of Theorem 3.5** Suppose that $X_t$ is conditionally symmetric around the conditional mean $\mu$ for all $t \geqslant 1$. This implies that for all $\theta \in \mathbb{R}^d$, $\langle \theta, X_t - \mu \rangle \sim -\langle \theta, X_t - \mu \rangle | \mathcal{F}_{t-1}$. de la Peña (1999, Lemma 1) (see also Howard et al., 2020, Lemma 3) shows that for conditionally symmetric random variables $(Y_t)_{t \geqslant 1}$ with mean 0, the process given by $\prod_{i \leqslant t} \exp\{\lambda_i Y_i - \lambda_i^2 Y_i^2 / 2\}$ is a nonnegative supermartingale. Consider this process with $Y_i = \langle \theta, X_i - \mu \rangle$, and applying Proposition 1.1 with prior $\rho_0$ and posteriors $\rho_\vartheta$, $\vartheta \in \mathbb{S}^{d-1}$, (with covariance $\beta^{-1} I_d$ as usual) we obtain that with probability $1 - \alpha$, simultaneously for all $t \geqslant 1$,

$$\int_{\mathbb{R}^d} \sum_{i \leqslant t} \lambda_i \langle \theta, X_i - \mu \rangle \rho_\vartheta(\mathrm{d}\theta) \leqslant \int_{\mathbb{R}^d} \sum_{i \leqslant t} \frac{\lambda_i^2}{2} \langle \theta, X_i - \mu \rangle^2 \rho_\vartheta(\mathrm{d}\theta) + \frac{\beta}{2} + \log(1/\alpha).$$

Then, as in the proof of Theorem 3.2 above, we note that

$$\int \langle \theta, X_i - \mu \rangle^2 \rho_\vartheta(\mathrm{d}\theta) \leqslant \left(1 + \frac{1}{\beta}\right)(\|X_i\| + v)^2.$$

Therefore, we obtain that with probability $1 - \alpha$, simultaneously for all $t \geqslant 1$,

$$\sup_{\vartheta \in \mathbb{S}^{d-1}} \sum_{i \leqslant t} \langle \vartheta, X_i - \mu \rangle \leqslant \left(1 + \frac{1}{\beta}\right) \sum_{i \leqslant t} \frac{\lambda_i^2}{2}(\|X_i\| + v)^2 + \frac{\beta}{2} + \log(1/\alpha).$$

Taking $\beta = 1$ completes the proof.

**Proof of Lemma 3.6** First, let us observe the relationship

$$0 \leqslant 1 - \frac{a \wedge 1}{a} \leqslant a, \quad \forall a > 0.$$

This is easily seen by case analysis. Indeed, for $a \geqslant 1$, we have $(a \wedge 1)/a = 1/a$ and $1 - 1/a \leqslant 1 \leqslant a$. For $a < 1$, we have $1 - (a \wedge 1)/a = 1 - 1 = 0 \leqslant a$.

Now, let

$$\alpha(X) = \frac{\lambda \|X\| \wedge 1}{\lambda \|X\|},$$

and note that the above analysis demonstrates that

$$|\alpha(X) - 1| = 1 - \alpha(X) \leqslant \lambda \|X\|. \tag{47}$$

Therefore,

$$
\begin{aligned}
\langle \vartheta, \mu_t - \mu \rangle &= \langle \vartheta, \mathbb{E}[\alpha(X)X] - \mathbb{E}[X] \rangle \\
&= \mathbb{E}(\alpha(X) - 1)\langle \vartheta, X \rangle \\
&\leqslant \mathbb{E}|\alpha(X) - 1||\langle \vartheta, X \rangle| \\
&\leqslant \mathbb{E}\lambda \|X\| \|\vartheta\| \|X\| && \text{by (47) and Cauchy-Schwarz} \\
&= \lambda \mathbb{E}\|X\|^2 && \|\vartheta\| = 1 \\
&\leqslant \lambda v^2 && \text{by assumption.}
\end{aligned}
$$

This proves the claim.

**Proof of Theorem 3.7** First let us state the PAC-Bayesian theorem upon which we rely. The following, due to Chugg et al. (2023, Corollary 15), is a time-uniform extension of the bound by Catoni (2004, Equation (5.2.1)). It is based on applying Proposition 1.1 to the supermartingale defined by $U_t(\theta) = \prod_{i \leqslant t} \exp\{\lambda_i f_i(\theta) - \log \mathbb{E} \exp(\lambda_i f_i(\theta))\}$.

**Lemma C.4.** *Let $(X_t)_{t \geqslant 1} \sim P$ and let $\{f_t : \mathcal{X} \times \Theta \to \mathbb{R}\}$ be a sequence of measurable functions. Fix a prior $\nu$ over $\Theta$. Then, with probability $1 - \alpha$ over $P$, for all $t \geqslant 1$ and all distributions $\rho$ over $\Theta$,*

$$\sum_{i \leqslant t} \int_\Theta f_i(X_i, \theta) \rho(\mathrm{d}\theta) \leqslant \sum_{i \leqslant t} \int_\Theta \log \mathbb{E}_{i-1} e^{f_i(X, \theta)} \rho(\mathrm{d}\theta) + D_{\mathrm{KL}}(\rho \| \nu) + \log \frac{1}{\alpha}.$$

As stated in Section 3, we apply Lemma C.4 with the functions $f_i(X_i, \theta) = \lambda_i \langle \theta, \mathsf{th}_i(X_i) - \mu_i^{\mathsf{th}} \rangle$. Keeping in mind that $\mathbb{E}_{\theta \sim \rho_\vartheta} \langle \theta, \mathsf{th}_i(X_i) - \mu_i^{\mathsf{th}} \rangle = \langle \vartheta, \mathsf{th}_i(X_i) - \mu_i^{\mathsf{th}} \rangle$, we obtain that with probability $1 - \alpha$, simultaneously for all $t \geqslant 1$,

$$\sup_{\vartheta \in \mathbb{S}^{d-1}} \sum_{i \leqslant t} \lambda_i \langle \vartheta, \mathsf{th}_i(X_i) - \mu_i^{\mathsf{th}} \rangle \leqslant \sum_{i \leqslant t} \mathop{\mathbb{E}}_{\theta \sim \rho_\vartheta} \log \mathbb{E} \left\{ \mathrm{e}^{\lambda_i \langle \theta, \mathsf{th}_i(X_i) - \mu_i^{\mathsf{th}} \rangle} | \mathcal{F}_{i-1} \right\} + \frac{\beta}{2} + \log \frac{1}{\alpha}. \tag{48}$$

Our second technical lemma, following Catoni and Giulini (2018), helps bound the right hand side of the above.

**Lemma C.5.** *For all $t \geqslant 1$,*

$$\mathop{\mathbb{E}}_{\theta \sim \rho_\vartheta} \log \mathop{\mathbb{E}}_{X \sim P} \left\{ \mathrm{e}^{\lambda_i \langle \theta, \mathsf{th}_i(X) - \mu_i^{\mathsf{th}} \rangle} | \mathcal{F}_{t-1} \right\} \leqslant \frac{1}{4} v^2 \lambda_i^2 e^{2/\beta + 2}.$$

*Proof.* To begin, notice that Jensen's inequality gives

$$\int \log \mathop{\mathbb{E}}_{X \sim P} \left\{ \mathrm{e}^{\lambda_i \langle \theta, \mathsf{th}_i(X) - \mu_i^{\mathsf{th}} \rangle} \right\} \rho_\vartheta(\mathrm{d}\theta)$$

$$\leqslant \log \int \mathop{\mathbb{E}}_{X \sim P} \left\{ \mathrm{e}^{\lambda_i \langle \theta, \mathsf{th}_i(X) - \mu_i^{\mathsf{th}} \rangle} \right\} \rho_\vartheta(\mathrm{d}\theta)$$

$$= \log \mathop{\mathbb{E}}_{X \sim P} \left\{ \int \mathrm{e}^{\lambda_i \langle \theta, \mathsf{th}_i(X) - \mu_i^{\mathsf{th}} \rangle} \rho_\vartheta(\mathrm{d}\theta) \right\}$$

$$= \log \mathbb{E} \exp \left( \lambda_i \langle \vartheta, \mathsf{th}_i(X) - \mu_i^{\mathsf{th}} \rangle + \frac{\lambda_i^2}{2\beta} \| \mathsf{th}_i(X) - \mu_i^{\mathsf{th}} \|^2 \right),$$

where the final line uses the usual closed-form expression of the multivariate Gaussian MGF. Define the functions on $\mathbb{R}$

$$g_1(x) := \frac{1}{x}(e^x - 1),$$

and

$$g_2(x) := \frac{2}{x^2}(e^x - x - 1),$$

(with $g_1(0) = g_2(0) = 1$ by continuous extension). Both $g_1$ and $g_2$ are increasing. Notice that

$$e^{x+y} = 1 + x + \frac{x^2}{2} g_2(x) + g_1(y) y e^x. \tag{49}$$

Consider setting $x$ and $y$ to be the two terms in the CGF above, i.e.,

$$x = \lambda_i \langle \vartheta, \mathsf{th}_i(X) - \mu_i^{\mathsf{th}} \rangle,$$

$$y = \frac{\lambda_i^2}{2\beta} \| \mathsf{th}_i(X) - \mu_i^{\mathsf{th}} \|^2,$$

where we recall that $\vartheta \in \mathbb{S}^{d-1}$. Before applying (49) we would like to develop upper bounds on $x$ and $y$. Observe that

$$\| \mathsf{th}_i(X) \| = \frac{\lambda_i \| X \| \wedge 1}{\lambda_i} \leqslant \frac{1}{\lambda_i},$$

and consequently,

$$\left\| \mu_i^{\mathsf{th}} \right\| = \| \mathbb{E} \mathsf{th}_i(X) \| \leqslant \mathbb{E} \| \mathsf{th}_i(X) \| \leqslant \frac{1}{\lambda_i},$$

by Jensen's inequality. Therefore, by Cauchy-Schwarz and the triangle inequality,

$$x \leqslant \lambda_i \| \vartheta \| \left\| \mathsf{th}_i(X) - \mu_i^{\mathsf{th}} \right\| \leqslant \lambda_i (\| \mathsf{th}_i(X) \| + \left\| \mu_i^{\mathsf{th}} \right\|) \leqslant 2.$$

Via similar reasoning, we can bound $y$ as

$$y \leqslant \frac{\lambda_i^2}{2\beta}(\|\mathsf{th}_i(X)\| + \|\mu_i^{\mathsf{th}}\|)^2 \leqslant \frac{2}{\beta}.$$

Finally, substituting in these values of $x$ and $y$ to (49), taking expectations, and using the fact that $g_1$ and $g_2$ are increasing gives

$$\mathbb{E}\left[\exp\left(\langle\vartheta, \mathsf{th}_i(X) - \mu_i^{\mathsf{th}}\rangle + \frac{1}{2\beta}\|\mathsf{th}_i(X) - \mu_i^{\mathsf{th}}\|^2\right)\bigg|\mathcal{F}_{i-1}\right]$$

$$\leqslant 1 + \lambda_i\mathbb{E}[\langle\vartheta, \mathsf{th}_i(X) - \mu_i^{\mathsf{th}}\rangle|\mathcal{F}_{i-1}] + g_2(2)\frac{\lambda_i^2}{2}\mathbb{E}[\langle\vartheta, \mathsf{th}_i(X) - \mu_i^{\mathsf{th}}\rangle^2|\mathcal{F}_{i-1}]$$

$$+ g_1\left(\frac{2}{\beta}\right)\frac{\lambda_i^2 e^2}{2\beta}\mathbb{E}[\|\mathsf{th}_i(X) - \mu_i^{\mathsf{th}}\|^2|\mathcal{F}_{i-1}]$$

$$\leqslant 1 + g_2(2)\frac{\lambda_i^2}{2}\mathbb{E}[\|\mathsf{th}_i(X) - \mu_i^{\mathsf{th}}\|^2|\mathcal{F}_{i-1}] + g_1\left(\frac{2}{\beta}\right)\frac{\lambda_i^2 e^2}{2\beta}\mathbb{E}[\|\mathsf{th}_i(X) - \mu_i^{\mathsf{th}}\|^2|\mathcal{F}_{i-1}],$$

where we've used that $\mathbb{E}[\mathsf{th}_i(X) - \mu_i^{\mathsf{th}}|\mathcal{F}_{i-1}] = 0$. Denote by $X'$ an iid copy of $X$. Using the notation $\mathbb{E}_{i-1}[\cdot] = \mathbb{E}[\cdot|\mathcal{F}_{i-1}]$, we can bound the norm as follows:

$$\mathbb{E}_{i-1}[\|\mathsf{th}_i(X) - \mu_i^{\mathsf{th}}\|^2] = \mathbb{E}_{i-1}[\|\mathsf{th}_i(X)\|^2] - \|\mu_i^{\mathsf{th}}\|^2$$

$$= \frac{1}{2}\mathbb{E}_{i-1}\left[\|\mathsf{th}_i(X)\|^2 - 2\langle\mathsf{th}_i(X), \mu_i^{\mathsf{th}}\rangle + \mathbb{E}_{i-1}[\|\mathsf{th}_i(X)\|^2]\right]$$

$$= \frac{1}{2}\mathbb{E}_X\left[\|\mathsf{th}_i(X)\|^2 - 2\left\langle\mathsf{th}_i(X), \mathbb{E}_{X'}[\mathsf{th}_i(X')|\mathcal{F}_{i-1}]\right\rangle\right.$$

$$\left.+ \mathbb{E}_{X'}\left[\|\mathsf{th}_i(X')\|^2|\mathcal{F}_{i-1}\right]\bigg|\mathcal{F}_{i-1}\right]$$

$$= \frac{1}{2}\mathbb{E}_{X,X'}\left[\|\mathsf{th}_i(X) - \mathsf{th}_i(X')\|^2|\mathcal{F}_{i-1}\right]$$

$$\leqslant \frac{1}{2}\mathbb{E}_{X,X'}\left[\|X - X'\|^2|\mathcal{F}_{i-1}\right]$$

$$= \mathbb{E}_{i-1}\|X - \mathbb{E}_{i-1}X\|^2 \leqslant \mathbb{E}_{i-1}\|X\|^2 \leqslant v^2,$$

where the first inequality uses the basic fact from convex analysis that, for a closed a convex set $D \subset \mathbb{R}^n$ and any $\mathbf{x}, \mathbf{y} \in \mathbb{R}^n$,

$$\|\Pi_D(\mathbf{x}) - \Pi_D(\mathbf{y})\| \leqslant \|\mathbf{x} - \mathbf{y}\|,$$

where $\Pi_D$ is the projection onto $D$. Putting everything together thus far, we have shown that

$$\int \log \mathbb{E}_{X \sim P}\left\{e^{\lambda_i\langle\theta, \mathsf{th}_i(X) - \mu_i^{\mathsf{th}}\rangle}\bigg|\mathcal{F}_{i-1}\right\}\rho_\vartheta(\mathrm{d}\theta)$$

$$\leqslant \log\left\{1 + g_2(2)\frac{\lambda_i^2 v^2}{2} + g_1\left(\frac{2}{\beta}\right)\frac{\lambda_i^2 e^2}{2\beta}v^2\right\}$$

$$\leqslant g_2(2)\frac{\lambda_i^2 v^2}{2} + g_1\left(\frac{2}{\beta}\right)\frac{\lambda_i^2 e^2}{2\beta}v^2$$

$$= v^2\frac{\lambda_i^2}{4}\left\{e^{2/\beta+2} - 3\right\}$$

$$\leqslant \frac{1}{4}v^2\lambda_i^2 e^{2/\beta+2},$$

which is the desired inequality. $\qquad\square$

To obtain the main result, we apply Lemmas C.5 and 3.6. For all $\vartheta \in \mathbb{S}^{d-1}$,

$$\sum_{i\leqslant t}\lambda_i\langle\vartheta, \mathsf{th}_i(X_i) - \mu\rangle = \sum_{i\leqslant t}\lambda_i(\langle\vartheta, \mathsf{th}_i(X_i) - \mu_i^{\mathsf{th}}\rangle + \langle\vartheta, \mu_i^{\mathsf{th}} - \mu\rangle)$$

$$
\leqslant \sum_{i \leqslant t} \lambda_i \langle \vartheta, \mathsf{th}_i(X_i) - \mu_i^{\mathsf{th}} \rangle + v^2 \sum_{i \leqslant t} \lambda_i^2
$$

$$
\leqslant \frac{v^2 e^{\frac{2}{\beta}+2}}{4} \sum_{i \leqslant t} \lambda_i^2 + v^2 \sum_{i \leqslant t} \lambda_i^2 + \frac{\beta}{2} + \log\left(\frac{1}{\alpha}\right)
$$

$$
\leqslant v^2 \left(2 e^{\frac{2}{\beta}+2} + 1\right) \sum_{i \leqslant t} \lambda_i^2 + \frac{\beta}{2} + \log\left(\frac{1}{\alpha}\right),
$$

Noting that

$$
\sup_{\vartheta \in \mathbb{S}^{d-1}} \sum_{i \leqslant t} \lambda_i \langle \vartheta, \mathsf{th}_i(X_i) - \mu \rangle = \left\| \sum_{i \leqslant t} \lambda_i (\mathsf{th}_i(X_i) - \mu) \right\|,
$$

and dividing through by $\sum_{i \leqslant t} \lambda_i$ gives the desired result.

## D   Simulation Details

Code can be found at https://github.com/bchugg/confidence-spheres.

Let us first describe the result of Duchi and Haque (2024), enabling us to transform any fixed-time estimator into a sequential estimator which loses only an iterated-logarithm factor. Let $\widehat{\mu}_n = \widehat{\mu}(X_1, \dots, X_n)$ be an estimator of the mean $\mu$ which satisfies the following deviation inequality for iid observations $X_1, \dots, X_n$: For all $n \geqslant 1$,

$$
\mathbb{P}(\|\widehat{\mu}_n - \mu\| \geqslant F(\log(1/\alpha), n)) \leqslant \alpha, \tag{50}
$$

for some function $F : \mathbb{R} \times \mathbb{N} \to (0, \infty)$. Then

$$
\Pr\big(\exists k : \|\widehat{\mu}_{2^k} - \mu\| \geqslant F(\log(\pi^2 k^2/6\alpha, 2^k))\big) \leqslant \alpha, \tag{51}
$$

implying that the estimator defined as

$$
\widehat{\mu}_t^{DH} = \begin{cases} \widehat{\mu}_t & \text{if } t = 2^k, \\ \widehat{\mu}_{t-1}, & \text{otherwise}, \end{cases}
$$

satisfies a time-uniform bound which suffers only an iterated logarithm penalty (plus some constants) over the original. We call this a "doubling strategy," as the estimator is updated every $2^k$ timesteps, $k \in \mathbb{N}$. In order to plot the boundary, at every time step $t$, if $t = 2^k$ for some $k$, we plot $F(\log(\pi^2 k^2/6\alpha, 2^k))$, and otherwise we plot the previous value of the boundary.

Let us now turn to the experimental details.

**Section 2.1.**   The bound in Theorem 2.2 is implemented with the parameters

$$
\beta = \sqrt{\frac{2\,\mathrm{Tr}(\Sigma) \log(1/\alpha)}{\|\Sigma\|}}, \quad \lambda_t = \sqrt{\frac{\beta + 2\log(1/\alpha)}{(\|\Sigma\| + \mathrm{Tr}(\Sigma)/\beta) t \log(t + 10e^4)}}.
$$

Theorem 2.3 is implemented directly as stated. We compute the bound of Hsu et al. (2012) as in (9), making it time-uniform via one of two methods: Either a union bound (taking $\alpha_t = \alpha/(t^2 + t)$ at each timestep $t$), or by the doubling method of Duchi and Haque (2024) above. In the left hand side of Figure 1 we use $\mathrm{Tr}(\Sigma^2) = \mathrm{Tr}(\Sigma) = \|\Sigma\| = 1$ (eg distributions with identity covariance matrix). In the figure on the right hand side, we fix $\|\Sigma\| = 1$, $\mathrm{Tr}(\Sigma^2) = 10$ and vary $\mathrm{Tr}(\Sigma)$.

**Section 3.3.**   For Theorem 3.7 we use $\beta = 4$ and

$$
\lambda_t = \sqrt{\frac{\log(1/\alpha)}{20v^2 t \log(t + 10e^4)}}.
$$

We use the following bound on the tournament median-of-means estimator: With probability $1 - \delta$,

$$\|\widehat{\mu}_n - \mu\| \leqslant \max\left\{240\sqrt{\frac{\lambda_{\max}\log(2/\delta)}{n}}, 960\sqrt{\frac{\text{Tr}(\Sigma)}{n}}\right\} \tag{52}$$

For the geometric median-of-means estimator, the tightest constants we could find come from the survey of Lugosi and Mendelson (2019a), which gives the following bound: With probability $1 - \delta$,

$$\|\widehat{\mu}_n - \mu\| \leqslant 4\sqrt{\frac{\text{Tr}(\Sigma)(8\log(1/\delta) + 1)}{n}}. \tag{53}$$

For the GMoM estimator we fix $\|\Sigma\| = 1$ and set $\text{Tr}(\Sigma) = 5$ and $v^2 = 5$ for all estimators. We set $\text{Tr}(\Sigma) = v^2$ for the reasons discussed in the caption, namely we are interested in how the bounds behave as functions of their multipliers on the know variance/second moment bound.

## E Empirical-Bernstein bound

Suppose there exists some $B \in \mathbb{R}_+$ such that $\sup_{X \in \mathcal{X}} \|X\| \leqslant B$. In this section we present an empirical-Bernstein bound, meaning a bound whose width adapts to the observations themselves, not on any *a priori* upper bound thereof like a traditional Hoeffding or Bernstein bound. The reason we present the bound in the appendix is that the width of the CSS is dimension-dependent. This is suboptimal as there exists a dimension-independent Bernstein bound for bounded random vectors (Gross, 2011; Kohler and Lucchi, 2017). Indeed, very recently, Martinez-Taboada and Ramdas (2024) gave a dimension-free empirical Bernstein bound in smooth Banach spaces (using different techniques). Whether a variational approach exists to providing a dimension-free empirical Bernstein bound remains an open question.

For any $t$, let $\bar{\mu}_t := \frac{1}{t}\sum_{i \leqslant t} X_i$ be the empirical mean at time $t$, and define the function $\psi_E(\lambda) := |\log(1 - \lambda) + \lambda|$ for $\lambda \in [0, 1)$. The following supermartingale is a multivariate analogue of that used by Waudby-Smith and Ramdas (2023, Eqn. (13)) to construct CSs, which in turn is based off of work by Howard et al. (2020; 2021) that generalized a lemma by Fan et al. (2015). The proof is in Appendix C.

**Lemma E.1.** *Suppose $(X_t)_{t \geqslant 1}$ satisfies $\sup_t \|X_t\| \leqslant B$ and Assumption 2. Let $(\lambda_t)_{t \geqslant 1}$ be a predictable sequence in $[0, 1)$. For each $\theta \in \mathbb{S}^{d-1}$, the process $N(\theta) \equiv (N_t(\theta))_{t \geqslant 1}$ is a nonnegative $P$-supermartingale, where*

$$N_t(\theta) = \prod_{i \leqslant t} \exp\left\{\frac{\lambda_i}{2B}\langle\theta, X_i - \mu\rangle - \frac{\psi_E(\lambda_i)}{(2B)^2}\langle\theta, X_i - \bar{\mu}_{i-1}\rangle^2\right\}.$$

*Proof.* Fan et al. (2015, Equation (4.11)) demonstrates that for all $v \geqslant -1$ and $\lambda \in [0, 1)$,

$$\exp\{\lambda v - \psi_E(\lambda)v^2\} \leqslant 1 + \lambda v, \tag{54}$$

where we recall that $\psi_E(\lambda) = |\log(1 - \lambda) + \lambda| = -\lambda - \log(1 - \lambda)$. In order to demonstrate that $(N_t(\theta))$ is a $P$-NSM, it suffices to demonstrate that

$$\mathbb{E}_P\left[\exp\left\{\frac{\lambda_t}{2B}\langle\theta, X_t - \mu\rangle - \frac{\psi_E(\lambda_t)}{(2B)^2}\langle\theta, X_t - \bar{\mu}_{t-1}\rangle^2\right\}\middle|\mathcal{F}_{t-1}\right] \leqslant 1,$$

for all $t \geqslant 1$. Inspired by a trick of Howard et al. (2021, Section A.8), set

$$Y_t := \frac{X_t - \mu}{2B}, \quad \delta_t = \frac{\bar{\mu}_t - \mu}{2B},$$

and observe that $Y_t - \delta_{t-1} = \frac{1}{2B}(X_t - \bar{\mu}_{t-1})$. Then,

$$\exp\left\{\frac{\lambda_t}{2B}\langle\theta, X_t - \mu\rangle - \frac{\psi_E(\lambda_t)}{(2B)^2}\langle\theta, X_t - \bar{\mu}_{t-1}\rangle^2\right\}$$
$$= \exp\left\{\lambda_t\langle\theta, Y_t\rangle - \psi_E(\lambda_t)\langle\theta, Y_t - \delta_{t-1}\rangle^2\right\}$$

$$= \exp\left\{\lambda_t \langle \theta, Y_t - \delta_{t-1}\rangle - \psi_E(\lambda_t)\langle\theta, Y_t - \delta_{t-1}\rangle^2\right\} \exp(\lambda_t \langle\theta, \delta_{t-1}\rangle).$$

Since $\|\theta\| = 1$ we have, by construction,

$$|\langle\theta, Y_t - \delta_{t-1}\rangle| \leqslant \frac{1}{2B}(\|X_t\| + \|\bar{\mu}_{t-1}\|) \leqslant 1,$$

since $\bar{\mu}_{t-1}$ is an average of vectors each with norm at most $B$. Therefore, we may apply (54) with $\upsilon = \langle\theta, Y_t - \delta_{t-1}\rangle$, which, paired with the display above yields

$$\exp\left\{\frac{\lambda_t}{2B}\langle\theta, X_t - \mu\rangle - \frac{\psi_E(\lambda_t)}{(2B)^2}\langle\theta, X_t - \bar{\mu}_{t-1}\rangle^2\right\} \leqslant (1 + \lambda_t\langle\theta, Y_t - \delta_{t-1}\rangle)\exp(\lambda_t\langle\theta, \delta_{t-1}\rangle).$$

Taking expectations and noticing that $\mathbb{E}_P[Y_t|\mathcal{F}_{t-1}] = 0$ gives

$$\mathbb{E}_P\left[\exp\left\{\frac{\lambda_t}{2B}\langle\theta, X_t - \mu\rangle - \frac{\psi_E(\lambda_t)}{(2B)^2}\langle\theta, X_t - \bar{\mu}_{t-1}\rangle^2\right\}\middle|\mathcal{F}_{t-1}\right]$$
$$\leqslant (1 - \lambda_t\langle\theta, \delta_{t-1}\rangle)\exp(\lambda_t\langle\theta, \delta_{t-1}\rangle) \leqslant 1,$$

where the final inequality uses the fact that $1 + x \leqslant e^x$ for all $x \in \mathbb{R}$. This completes the proof. $\square$

The "$-\bar{\mu}_{i-1}$" can be replaced by any predictable estimate, and is a particularly important ingredient which even in the scalar setting does not immediately follow from Fan et al. (2015), but it critically changes the bound's dependence from $\mathbb{E}[\|X\|^2]$ to $\mathbb{E}[\|X - \mu\|^2]$. Applying Proposition 1.1 to the supermartingale defined above and using uniform distributions as our priors and posteriors, we obtain the following.

**Theorem E.2.** *Suppose $(X_t)_{t\geqslant 1} \sim P$ for any $P$ obeying $\sup_{X\sim P}\|X\| \leqslant 1/2$ and Assumption 2. Let $(\lambda_t)_{t\geqslant 1}$ be a predictable sequence in $(0,1)$ and fix any $0 < \epsilon < 1$. Then, for all $\alpha \in (0,1)$, with probability at least $1 - \alpha$, simultaneously for all $t \geqslant 1$,*

$$\left\|\frac{\sum_{i\leqslant t}\lambda_i X_i}{\sum_{i\leqslant t}\lambda_i} - \mu\right\| \leqslant \frac{\sum_{i\leqslant t}\psi_E(\lambda_i)\|X_i - \bar{\mu}_{i-1}\|^2 + d\log(1/\epsilon) + \log\frac{1}{\alpha}}{(1-\epsilon)\sum_{i\leqslant t}\lambda_i}. \tag{55}$$

*Proof.* As in Section 2.3, let $\rho_\vartheta$ be a uniform distribution over $\mathbb{B}^d$ with radius $\epsilon$ centered at $\vartheta \in (1-\epsilon)\mathbb{S}^{d-1}$ and let $\nu$ be the uniform distribution over $\mathbb{S}^{d-1}$. (We note that we can consider uniform distributions over the ball because $\psi_E$ is super-Gaussian; see the discussion in Section 2.3 and Lemma 2.10.) Applying Proposition 1.1 to $N_t(\theta)$ from Lemma E.1 gives that with probability $1 - \alpha$, for all $t \geqslant 1$ and all $\vartheta \in (1-\epsilon)\mathbb{S}^{d-1}$,

$$\sum_{i\leqslant t}\lambda_i\langle\vartheta, X_i - \mu\rangle = \int_{\mathbb{S}^{d-1}}\sum_{i\leqslant t}\lambda_i\langle\theta, X_i - \mu\rangle\rho_\vartheta(\mathrm{d}\theta)$$

$$\leqslant \int_{\mathbb{S}^{d-1}}\sum_{i\leqslant t}\psi_E(\lambda_i)\langle\theta, X_i - \bar{\mu}_{i-1}\rangle^2\rho_\vartheta(\mathrm{d}\theta) + D_{\mathrm{KL}}(\rho_\vartheta\|\nu) + \log(1/\alpha)$$

$$\leqslant \int_{\mathbb{S}^{d-1}}\sum_{i\leqslant t}\psi_E(\lambda_i)\|X_i - \bar{\mu}_{i-1}\|^2\rho_\vartheta(\mathrm{d}\theta) + d\log(1/\epsilon) + \log(1/\alpha)$$

$$= \sum_{i\leqslant t}\psi_E(\lambda_i)\|X_i - \bar{\mu}_{i-1}\|^2 + d\log(1/\epsilon) + \log(1/\alpha).$$

That is, with probability $1 - \alpha$, for all $t \geqslant 1$,

$$\sup_{\vartheta\in(1-\epsilon)\mathbb{S}^{d-1}}\left\langle\vartheta, \sum_{i\leqslant t}(X_i - \mu)\right\rangle \leqslant \sum_{i\leqslant t}\psi_E(\lambda_i)\|X_i - \bar{\mu}_{i-1}\|^2 + d\log(1/\epsilon) + \log(1/\alpha).$$

The left hand side is equal to $(1-\epsilon)\left\|\sum_{i\leqslant t}\lambda_i(X_i - \mu)\right\|$. Dividing by $\sum_i\lambda_i$ results in the bound:

$$\left\|\frac{\sum_{i\leqslant t}\lambda_i X_i}{\sum_{i\leqslant t}\lambda_i} - \mu\right\| \leqslant \frac{\sum_{i\leqslant t}\psi_E(\lambda_i)\|X_i - \bar{\mu}_{i-1}\|^2 + d\log(1/\epsilon) + \log(1/\alpha)}{(1-\epsilon)\sum_{i\leqslant t}\lambda_i},$$

which is the desired result. $\square$

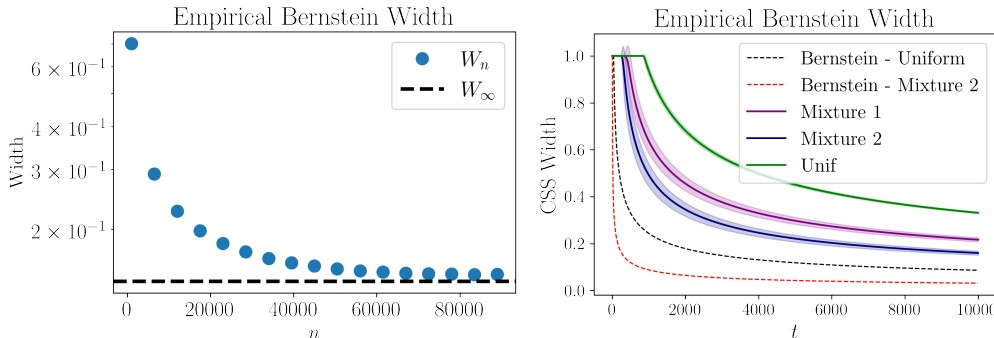

Figure 3: **Left:** The width of our empirical Bernstein CI as $n \to \infty$, which approaches its asymptotic width $W_\infty$. We use $\alpha = 0.05$, $d = 2$, and random vectors comprised of two Beta(10,10) distributed random variables. **Right:** Performance of our empirical Bernstein bound compared to the multivariate (non-empirical) Bernstein bound baseline, with oracle access to the true variance. Shaded areas provide the standard deviation across 100 trials. The distributions are mixtures of betas and binomials. Mixture 2 has the lowest variance. As the variance decreases our empirical bounds get tighter and approach the tightest known oracle bounds (black and red dotted lines) (Gross, 2011; Kohler and Lucchi, 2017).

Theorem E.2 immediately yields empirical-Bernstein CIs when instantiated at a fixed time $n$. For intuition, note that $\psi_E(\lambda) \asymp \lambda^2/2$ for small $\lambda$, so for $\lambda_i \propto 1/\sqrt{n}$ for all $i$, the denominator is $\sqrt{n}$ while the numerator's first term does not depend on $n$. More interesting though is the dependence on the variance. Below we analyze the asymptotic width of the CI under iid data when taking $\lambda_i = \sqrt{2d \log(1/\epsilon) + 2 \log(1/\alpha)/\widehat{\sigma}^2_{i-1} n}$, where $\widehat{\sigma}^2_n = \frac{1}{n} \sum_{i \leqslant n} \|X_i - \bar{\mu}_i\|^2$ is an empirical estimate of the variance $\sigma^2 = \mathbb{V}(X) = \mathbb{E}[\|X - \mu\|^2]$. The width $W_n$ (the RHS of (55)) obeys

$$\sqrt{n} W_n \xrightarrow{a.s.} \frac{\sigma}{1 - \epsilon} \sqrt{2d \log(1/\epsilon) + 2 \log(1/\alpha)}. \tag{56}$$

Therefore, the width of the bound scales with the true unknown variance. This can be seen as a generalization of known results in $d = 1$ to multivariate settings. Indeed, in the scalar setting, the empirical-Bernstein bound of Waudby-Smith and Ramdas (2023, Theorem 2) gives an asymptotic width of $\sigma \sqrt{2 \log(2/\alpha)}$, and that of Maurer and Pontil (2009) scales as $\sigma \sqrt{2 \log(4/\alpha)}$. We might also compare our result to *oracle* Bernstein bounds in the multivariate setting, which assume knowledge of the variance. Kohler and Lucchi (2017), based on previous work by Gross (2011), show that if $(X_t)^n_{t=1}$ are iid, then with probability $1 - \alpha$, $\left\| \frac{1}{n} \sum_{i \leqslant n} X_i - \mu \right\| \leqslant \sigma \sqrt{8(\log(1/\alpha)+1/4)/n}$. Note that this a bound has no dimension-dependence.

In the sequential setting, applying this result with $\lambda_t \asymp \sqrt{d + \log(1/\alpha)/t \log t}$ results in a width scaling as $\widetilde{O}(B \sqrt{d + \log(1/\alpha) \log(t)/t})$, where $\widetilde{O}$ hides iterated logarithm factors. Combined with the insights in the fixed-time setting explored above, we suggest taking $\lambda_t \asymp \sqrt{d + \log(1/\alpha)/\widehat{\sigma}^2_{t-1} t \log t}$. The extra $\log(t)$ in the denominator is required to ensure the rate is $\sqrt{\log(t)/t}$. We emphasize that our bounds hold for all predictable sequences $(\lambda_t)$ in $(0, 1)$, but the precise selection matters for the asymptotic width and the empirical performance.

## E.1 Asymptotic width

Here we study the asymptotic width of our empirical Bernstein confidence intervals and demonstrate that they scale with the true variance over time.

We assume the data are iid. Let $\sigma^2 = \mathbb{V}(X) = \mathbb{E}[\|X - \mu\|^2]$. Fix a sample size $n$, let $\widehat{\sigma}^2_n = \frac{1}{n} \sum^n_{i=1} \|X_i - \bar{\mu}_{i-1}\|^2$, and consider

$$\lambda_t = \sqrt{\frac{c(d \log(1/\epsilon) + \log(1/\alpha))}{\widehat{\sigma}^2_{t-1} n}}, \quad \forall t \geqslant 1,$$

for some constant $c$. Here we define $\widehat{\sigma}_0^2 = 1$. Our goal is to show that there exists a $c$ such that $\sqrt{n}W_n$, where $W_n$ is the width of the empirical-Bernstein interval in Theorem E.2, converges almost surely to the quantity $\frac{\sigma}{1-\epsilon}\sqrt{2d\log(1/\epsilon) + 2\log(1/\alpha)}$. Our proof follows similar steps to Waudby-Smith and Ramdas (2023, Appendix E.2) who prove the result in the scalar setting. Most of the relevant mechanics still go through, however.

We build up to the result via a sequence of lemmas.

**Lemma E.3.** $\widehat{\sigma}_n^2$ *converges to $\sigma^2$ almost surely.*

*Proof.* Decompose $\widehat{\sigma}_n^2$ as follows:

$$\widehat{\sigma}_n^2 = \frac{1}{n}\sum_{i=1}^{n}\|X_i - \bar{\mu}_{i-1}\|^2 = \frac{1}{n}\sum_{i=1}^{n}\|X_i - \mu + \mu - \bar{\mu}_{i-1}\|^2$$

$$= \frac{1}{n}\sum_{i=1}^{n}\|X_i - \mu\|^2 + \frac{2}{n}\sum_{i=1}^{n}\langle X_i - \mu, \mu - \bar{\mu}_{i-1}\rangle + \frac{1}{n}\sum_{i=1}^{n}\|\mu - \bar{\mu}_{i-1}\|^2. \tag{57}$$

Now, by the SLLN, the first sum converges to $\sigma^2$ almost surely. As for the second sum, write

$$\left|\frac{2}{n}\sum_{i=1}^{n}\langle X_i - \mu, \mu - \bar{\mu}_{i-1}\rangle\right| \leqslant \frac{2}{n}\sum_{i=1}^{n}\|X_i - \mu\|\|\mu - \bar{\mu}_{i-1}\| \leqslant \frac{2}{n}\sum_{i=1}^{n}\|\mu - \bar{\mu}_{i-1}\| \xrightarrow{a.s.} 0,$$

where almost sure convergence follows from combining the following three observations: (i) If the absolute value of a sequence converges to zero, then the sequence converges to zero; (ii) If a sequence converges to a value, then its partial sums converge to that value; (iii) $\bar{\mu}_{i-1}$ converges to $\mu$ a.s., so by continuous mapping theorem $\|\mu - \bar{\mu}_{i-1}\| \xrightarrow{a.s.} 0$. The third sum in (57) converges almost surely to 0 for the same reason. This completes the proof. $\square$

**Lemma E.4.** $\sum_{i\leqslant n}\psi_E(\lambda_i)\|X_i - \bar{\mu}_{i-1}\|^2$ *converges to $\frac{c}{2}(d\log(1/\epsilon) + \log(1/\alpha))$ almost surely.*

*Proof.* Define the function $\psi_H(\lambda_i) = \lambda_i^2/8$. The "H" stands for Hoeffding, and the notation is borrowed from Howard et al. (2020). Now, Waudby-Smith and Ramdas (2023) demonstrate that $\frac{\psi_E(\lambda)}{4\psi_H(\lambda)} \to 1$ as $\lambda \to 0$. Therefore, we can write $\psi_E(\lambda_i)/\psi_H(\lambda_i) = 4 + 4v_i$ where $v_i \to 0$ as $i \to \infty$, since $\lambda_i \xrightarrow{a.s.} 0$. Furthermore, by Lemma E.3, we can write $\sigma^2/\widehat{\sigma}_n^2 = 1 + u_n$ for some $u_n \xrightarrow{a.s.} 0$. Therefore,

$$\sum_{i=1}^{n}\psi_E(\lambda_i)\|X_i - \bar{\mu}_{i-1}\|^2 = \sum_{i=1}^{n}\frac{\psi_E(\lambda_i)}{\psi_H(\lambda_i)}\psi_H(\lambda_i)\|X_i - \bar{\mu}_{i-1}\|^2$$

$$= \frac{1}{2}\sum_{i=1}^{n}\lambda_i^2\|X_i - \bar{\mu}_{i-1}\|^2(1 + v_i)$$

$$= \frac{c(d\log(1/\epsilon) + \log(1/\alpha))}{2n}\sum_{i=1}^{n}\frac{\|X_i - \bar{\mu}_{i-1}\|^2}{\widehat{\sigma}_i^2}(1 + v_i)$$

$$= \underbrace{\frac{c(d\log(1/\epsilon) + \log(1/\alpha))}{2n}\sum_{i=1}^{n}\frac{\|X_i - \bar{\mu}_{i-1}\|^2}{\sigma^2}(1 + u_i)(1 + v_i)}_{:=K_i}.$$

Write $(1 + u_i)(1 + v_i) = 1 + w_i$ where $w_i \xrightarrow{a.s.} 0$. Then, the above sum becomes

$$\sum_{i=1}^{n}K_i(1 + w_i) = \frac{c(d\log(1/\epsilon) + \log(1/\alpha))}{2\sigma^2}\left\{\frac{1}{n}\sum_{i=1}^{n}\|X_i - \bar{\mu}_{i-1}\|^2(1 + w_i)\right\}$$

$$= \frac{c(d\log(1/\epsilon) + \log(1/\alpha))}{2\sigma^2}\left\{\widehat{\sigma}_n^2 + \widehat{\sigma}_n^2 w_i\right\}$$

$$\xrightarrow{a.s.} \frac{c(d\log(1/\epsilon) + \log(1/\alpha))}{2},$$

since $\widehat{\sigma}_n^2 \xrightarrow{a.s.} \sigma^2$ by Lemma E.3 and $w_i \xrightarrow{a.s.} 0$.

$\square$

**Lemma E.5.** $\frac{1}{\sqrt{n}}\sum_{i\leqslant n}\lambda_i$ *converges to* $\sqrt{\frac{c(d\log(1/\epsilon)+\log(1/\alpha))}{\sigma^2}}$ *almost surely.*

*Proof.* Via similar reasoning as above, we can write $\sigma/\widehat{\sigma}_i \to 1 + u_i$ for some $u_i \xrightarrow{a.s.} 0$. (Here we've employed the continuous mapping theorem on Lemma E.3.) Therefore,

$$\begin{aligned}
\frac{1}{\sqrt{n}}\sum_{i=1}^{n}\lambda_i &= \frac{1}{\sqrt{n}}\sum_{i=1}^{n}\sqrt{\frac{c(d\log(1/\epsilon)+\log(1/\alpha))}{\widehat{\sigma}_{i-1}^2 n}} \\
&= \frac{\sqrt{c(d\log(1/\epsilon)+\log(1/\alpha))}}{n\sigma}\sum_{i=1}^{n}\sqrt{\frac{\sigma^2}{\widehat{\sigma}_{i-1}^2}} \\
&\xrightarrow{a.s.} \sqrt{\frac{c(d\log(1/\epsilon)+\log(1/\alpha))}{\sigma^2}},
\end{aligned}$$

where we used the fact that if a sequence converges to a value then so too do its partial sums. $\square$

Now, let

$$W_n := \frac{\sum_{i\leqslant t}\psi_E(\lambda_i)\|X_i - \bar{\mu}_{i-1}\|^2}{(1-\epsilon)\sum_{i\leqslant t}\lambda_i} + \frac{d\log(1/\epsilon) + \log\frac{1}{\alpha}}{(1-\epsilon)\sum_{i\leqslant t}\lambda_i}, \tag{58}$$

be the width of our CI in Theorem E.2. We have

$$\begin{aligned}
\sqrt{n}W_n &= \frac{\sum_{i\leqslant t}\psi_E(\lambda_i)\|X_i - \bar{\mu}_{i-1}\|^2 + d\log(1/\epsilon) + \log\frac{1}{\alpha}}{\frac{1}{\sqrt{n}}(1-\epsilon)\sum_{i\leqslant t}\lambda_i} \\
&\xrightarrow{a.s.} \frac{\frac{c}{2}(d\log(1/\epsilon)+\log(1/\alpha)) + d\log(1/\epsilon) + \log(1/\alpha)}{(1-\epsilon)\sqrt{c(d\log(1/\epsilon)+\log(1/\alpha))/\sigma^2}} \\
&= \frac{\sigma\sqrt{d\log(1/\epsilon)+\log(1/\alpha)}}{1-\epsilon}\left(\frac{\sqrt{c}}{2} + \frac{1}{\sqrt{c}}\right).
\end{aligned}$$

Minimizing over $c$ gives $c = 2$, in which case we obtain

$$\sqrt{n}W_n \xrightarrow{a.s.} \frac{\sigma}{1-\epsilon}\sqrt{2d\log(1/\epsilon) + \log(1/\alpha)}.$$

