# OpenReview forum: "Time-Uniform Confidence Spheres for Means of Random Vectors"
_TMLR — Accepted by TMLR_

### Review · Reviewer_MUwX · 2025-04-02

**Summary Of Contributions:**

This paper confidence sequences for means of random vectors. As opposed to confidence intervals, confidence sequences hold uniformly over time. As the study is done in a multivariate setting, these sequences take the form of confidence sphere, ie, $P(\forall t,~ \Vert \hat{\mu} - \mu \Vert \leq c_t) \geq 1 - \alpha$.
This work is motivated by the fact that few results exist in 1D and by potential applications to bandits and stochastic optimization.
Inspired by previous work, the authors use a PAC-Bayesian approach of the problem proposed by Catoni and Giulini, which they extend by using the recently proposed uniform-in-time bounds of (Chugg et al.) which they nicely combine with the variational representation of the norm as $\Vert x \Vert = \sup_{\theta \in S^{d-1}} \langle \theta, x \rangle$ through the uniformity of PAC-Bayesian bounds over the posterior distribution. This allows them to derive explicit bounds in four main settings listed below. It is to be noted that most of the result are obtained under the assumption of conditionally constant mean vector.

- Subgaussian, log-concave, sub-$\psi$ bounds: Under a conditional subgaussian assumption with adapted covariance several confidence bounds around a linear estimator of the mean. These result recover existing bounds up to an absolute constant factor but are in addition time-uniform and adapted to sequences and martingale dependence.
The authors then study two possible relaxations of the subgaussian assumption through log-concave and the so-called sub-$\psi$ conditions and again recover existing results as particular cases of their theorems.

- Extension to time-varying means: The authors extend the so-called method of mixtures to handle the case where the conditional mean of the random vectors is also time-varying. This is a very nice extension of the results although it might suffer from the same curse of dimensionality as the sub-$\psi$ case.

- An extension from spehre to ellipsoidal confidence is briefly discussed.

- Heavy-tailed bounds: Finally, the authors prove several bounds in the case where only a finite second-order moment is assumed. The proposed bounds are dimension-free and again extend known results.

**Audience:**

Yes

**Claims And Evidence:**

Yes

**Requested Changes:**

- In order to improve the readability of the paper, I would strongly suggest including to the main text a table to compare some of your main results with existing results in the literature. Indeed, for a non-expert of confidence bounds, it is hard to grasp which results are completely new and which are less original.
- Similarly, I believe the large number of results and variants of the results might render the paper a bit hard to read, so maybe some results (like the instantiation at $t=n$ or some stitching results) could be only discussed in the appendix, or mentioned through a table in the main text. Similarly, the discussion of all the possible cases of sub-$\psi$ distributions page 10 might be moved to the appendix.

Regarding the theoretical claims and proofs, I noticed the following imprecisions, which should be addressed:
- Can you detail more the proof of Lemma 2.6, I am not sure to understand the lower bound on $H_n$, unless I misunderstood the notations. The same remark apply to the bound on the function $h_\Sigma(u)$.
- The notion of uniform distribution with radius $r$ is not clearly defined, you should mention that you mean uniform over a ball.
- There is a confusion between balls and spheres. For instance in section 1, the notation $S^{d-1}$ refers to the sphere of radius 1 while in equation 16 the same notation is clearly used for a ball (otherwise there is no absolute continuity). Moreover I believe the formula for the volume is wrong and $\pi^d$ should be replaced by $\pi^{d/2}$.
- Proof of them 3.5: there is a square missing in the second line (first term), it seems to be just a typo.
- Proof of Lemma 3.6: i think you don't need to invoke Jensen and just use the assumption of bounded variance.
- Do you need to use 2 pages to reprove Lemma B.3? You mention that it follows from existing work.
- Proof of them 2.8: there is a confusion in the definition of $\bar{\rho}_u$, is it truncated or not? Words are missing in the first sentences of this proof. In the KL computation, why is the last step an inequality (it is equal)? In the second line of the Kl estimation, there is a minus sign missing in the integrated. At the end of the proof there is a conflict of notation with $u$.

**Strengths And Weaknesses:**

**Strengths**

- The paper is quite exhaustive in terms of the different settings it covers.
- I looked at the proof of a good part of the main results (apart from appendix A and some of the variants) and it seems to be sound as far as I understand. However, there are a few imprecisions (see the requested changes section.)
- The main proof technique, based on the uniform-in-time PAC-Bayesian bounds, is very interesting. The use of these relatively new PAC-Bayesian bounds in this context is quite ingenious. Can you maybe discuss whether you are the first to use the theorems of (chugg et al.) for confidence bounds?
- The heavy-tailed bounds seems to be the first such bounds in a multivariate settings.


**Weaknesses**

- The proof technique for the sub-$\psi$ setting involves dealing with uniform distributions, which induces a dimension dependence of the proposed bounds. However, this seems to be also the case in the literature in such settings.
- Some results suffer from the curse of dimensionality, even though it seems that this is sometimes impossible to avoid.
- There are a lot of different settings and results, which might lose the reader a bit and harms the readability.

---

> ### Author Response · Authors · 2025-04-11
> **comment**
>
> We thank the reviewer for their close reading of the paper, and we're pleased that you find it interesting and exhaustive! We've uploaded a new version of the paper with changes in blue that address your comments. We'll also address them below.
>
> **Strengths and weaknesses**
>
> We agree it's disappointing that the bounds for general sub-$\psi$ processes have dimension-dependence. This appears to be unavoidable with our techniques, but it remains an open question whether different techniques can remove the dependence. The only other work studying general sub-$\psi$ processes in $\Re^d$ is Whitehouse et al. (2023), who also have dimension-dependence in their bounds (indeed, they always have dimension-dependence, even in the sub-Gaussian and sub-exponential cases, which our techniques avoid).
>
> **Requested Changes**
>
> 1. We've added Table 2, which provides further information on the relationship between our results and those found in the literature.
> 2. We've added Appendix A, which contains results that we've removed from the main body. This contains most of the fixed-time optimization results. However, we've kept in the fixed-time optimization of our sub-Gaussian bound in the main text as that is the main comparison for Hsu et al's bound. We hope the tables at the beginning the paper allow readers to find the section of interest to them, but we are open to moving even more to the appendix if needed. Alternatively, we are also happy to add a table of contents to the paper, though we're not sure if this is allowed in the TMLR format.
> 3. We've added more detail to the proof of Lemma 2.6.
> 4. We've clarified the notation with respect to balls and spheres, and now clearly differentiate the two.
> 5. Thank you for noticing the discrepancy between balls and spheres! It turns out that we were assuming that $\psi$ was *super-Gaussian*, which allowed us use balls instead of spheres. We've now clarified that Theorem 2.11 applies only to super-Gaussian $\psi$. For non super-Gaussian $\psi$, we can obtain a result using the von Mises-Fisher distribution. This has slightly worse dimension-dependence and constants, but has a similar overall form. We've added this result to Appendix A.3 and have added more discussion about the super-Gaussian assumption to the main paper.
> 6. The typo in the proof of Thm 3.5 has been fixed.
> 7. You're right - this is by assumption, Jensen was unecessary. Fixed.
> 8. We felt more comfortable keeping the proof of Lemma B.1 originally, but to assure ourselves of its correctness and also to keep readers from having to hunt down other references. But we're happy to omit it if the reviewer feels it's unecessary.
> 9. We've fixed the ambiguities in the proof.

---

### Review · Reviewer_VL2T · 2025-04-04

**Summary Of Contributions:**

This paper studies the problem of estimating the mean of multivariate data in a sequential manner and demonstrates a general framework to derive the nonparametric, time-uniform confidence sequences under martingale dependence using PAC-Bayesian techniques.

The main contribution is the application and extension of PAC-Bayesian techniques with the recent results on time-uniform bounds, which allows to derive the confidence sphere sequences under different distributional assumptions through choices of the specific martingale construction and family of posteriors thus weakening the iid data requirement to the martingale dependency.

**Audience:**

Yes

**Broader Impact Concerns:**

There is no concern here.

**Claims And Evidence:**

Yes

**Requested Changes:**

Should it be "The bound in (8) is therefore somewhat slightly looser" instead of "The bound in (5)" in the 4th to the last line on Page 7?

**Strengths And Weaknesses:**

This paper is well written, and the description in Section 1.2/1.3 has enhanced the overall presentation and also clearly demonstrated the main contribution. This work provided results for various important distribution classes, including sub-Gaussian, log-concave, general light tailed, and heavy-tailed distributions, and many of the derived confidence sphere sequences are shown to be favorably comparable with existing ones under different situations. The technique also seems to be quite general and different choices of the martingale and posterior might influence the final bound's tightness and complexity.

---

> ### Author Response · Authors · 2025-04-11
> **comment**
>
> We thank the reviewer for their positive comments on the paper! We're glad the exposition in Sections 1.2 and 1.3 is helpful. We're aware the paper is long so we're pleased that the introduction is clear. We also agree that the technique is quite general, and we're excited to see how far it can be pushed in the future.
>
> We've fixed the issue you highlighted -- it should have been "the bound in (7) is ...". The newest version of the paper reflects the change.

---

### Review · Reviewer_qnnE · 2025-04-09

**Summary Of Contributions:**

The authors constructed confidence sphere sequences (CSSs) based on PAC-Bayes theory in non-parametric settings. The results are dimension free and include both sub-Gaussian and heavy tailed multivariate distributions.

**Audience:**

Yes

**Broader Impact Concerns:**

NA (the paper is of theoretical nature).

**Claims And Evidence:**

Yes

**Requested Changes:**

Proposition 1.1. There is no Theorem 1 in Chugg et al 2023. And there is no proof of proposition 1.1 in the appendix.

Introduction, mention applications of CSSs and its background in machine learning.

Most theorems are probabilistic bounds, are there any related deterministic bounds?

The last paragraph on page 4. Give an example of applying proposition 1.1 and get a CSS.

**Strengths And Weaknesses:**

Pros:
- A list of confidence sphere sequences (CSSs) in different settings, based on a common framework.
- Good clarity and well organized results.

Cons:
- I could not identify any notable weaknesses. See below for minor remarks.

---

> ### Author Response · Authors · 2025-04-11
> **comment**
>
> We appreciate the reviewer's positive sentiments about the paper. We have fixed the various issues you highlighted, including:
>
> 1. Proposition comes from Theorem 4 in Chugg et al. (2023), not Theorem 1 as we had written. It seems we were citing an older arxiv version. We're now citing the JMLR version.
> 2. We've added several applications of CSSs to the introduction, agumenting the previous discussing of their use in A/B testing and adaptive experimentation.
> 3. Whether there are corresponding deterministic bounds is an interesting question. On one hand, the PAC-Bayesian technique we use (Prop 1.1) is fundamentally probabilistic. However, on the other hand, tail inequalities often have a corresponding deterministic _regret_ inequality ([Rakhlin and Sridharan 2017](https://proceedings.mlr.press/v65/rakhlin17a/rakhlin17a.pdf)). It would be interesting to explore that equivalence in this setting in followup work.
> 4. We have added a short paragraph to the bottom of Section 1.2 with an example of how to obtain a CSS in the case of iid 1-subGaussian random vectors.
>
> The changes can be seen in the newest version of the paper, where they are highlighted in blue.

---

### Decision · Action_Editor_PARz · 2025-05-01

**Recommendation:** Accept as is

**Comment:**

This paper astutely uses time-uniform PAC-Bayesian bounds to the problem of confidence sphere estimation. The authors prove results including for heavy-tailed random vectors. This should attract some interest in the community. Congratulations on a fine piece of work.

**Audience:**

I concur with the reviewers that this paper will be of interest to the ML community.

**Claims And Evidence:**

All claims are supported by appropriate evidence.